# Matrix directs trophoblast differentiation in a bioprinted organoid model of early placental development

Claire Richards [1,2], Hao Chen[3], Matthew O'Rourke[3], Ashley Bannister[1], Grace Owen[1], Alexander Volkerling[4], Arnab Ghosh[5,6], Catherine A. Gorrie[1], David Gallego-Ortega [7,8,9], Amy L. Bottomley [10], Matthew P. Padula [1], Kristine C. McGrath [1], Louise Cole[10], Philip M. Hansbro [3] & Lana McClements [1,2] ✉

Trophoblast organoids can provide crucial insights into mechanisms of placentation, however their potential is limited by highly variable extracellular matrices unable to reflect in vivo tissues. Here, we present a bioprinted placental organoid model, generated using the first trimester trophoblast cell line, ACH-3P, and a synthetic polyethylene glycol (PEG) matrix. Bioprinted or Matrigel-embedded organoids differentiate spontaneously from cytotrophoblasts into two major subtypes: extravillous trophoblasts (EVTs) and syncytiotrophoblasts (STBs). Bioprinted organoids are driven towards EVT differentiation and show close similarity with early human placenta or primary trophoblast organoids. Inflammation inhibits proliferation and STBs within bioprinted organoids, which aspirin or metformin (0.5 mM) cannot rescue. We reverse the inside-out architecture of ACH-3P organoids by suspension culture with STBs forming on the outer layer of organoids, reflecting placental tissue. Our bioprinted methodology is applicable to trophoblast stem cells. We present a high-throughput, automated, and tuneable trophoblast organoid model that reproducibly mimics the placental microenvironment in health and disease.

Organoids derived from single cells offer a powerful model for studying organ-like structures that recapitulate key features of native tissues[1]. However, their widespread reliance on animal-derived matrices such as Matrigel introduces variability and limits reproducibility, hindering biological insights and translational potential[2,3]. Given the critical role of the extracellular matrix (ECM) in guiding cell behaviour, including differentiation, organisation and function, there is a pressing need for tuneable, biologically relevant alternatives[4–8]. Emerging

[1]School of Life Sciences, Faculty of Science, University of Technology Sydney, Sydney, NSW, Australia. [2]Institute for Biomedical Materials and Devices, Faculty of Science, University of Technology Sydney, Sydney, NSW, Australia. [3]Centre for Inflammation, Centenary Institute and University of Technology Sydney, Faculty of Science, School of Life Sciences, Sydney, NSW, Australia. [4]Inventia Life Science Pty Ltd, Sydney, NSW, Australia. [5]School of Biomedical Sciences and Pharmacy, University of Newcastle, Callaghan, University of Newcastle, Callaghan, Newcastle, NSW, Australia. [6]Centre for Drug Repurposing and Medicines Research Program, Hunter Medical Research Institute, New Lambton Heights, Newcastle, NSW, Australia. [7]Single Cell Genomics Facility, School of Biomedical Engineering, Faculty of Engineering and Information Technology, University of Technology Sydney, Sydney, NSW, Australia. [8]Garvan Institute of Medical Research, The Kinghorn Cancer Centre, Darlinghurst, NSW, Australia. [9]School of Clinical Medicine, Faculty of Medicine, University of New South Wales Sydney, Kensington, NSW, Australia. [10]Microbial Imaging Facility at the Australian Institute for Microbial Imaging, Faculty of Science, University of Technology Sydney, Sydney, NSW, Australia. ✉e-mail: lana.mcclements@uts.edu.au

bioprinting technologies, including extrusion and droplet-based methods, enable precise deposition of cells within natural or synthetic hydrogels, offering scalable and reproducible platforms for 3D cell culture that are capable of supporting high-throughput workflows[4]. The placenta plays a vital role in supporting foetal development, and its dysfunction is implicated in numerous pregnancy complications, including preeclampsia, intrauterine growth restriction (IUGR), miscarriage, preterm birth, stillbirth, placental accreta, increta and percreta[5–14]. These conditions not only affect pregnancy outcomes but also have long-term health implications for both mother and child, increasing the risk of future cardiovascular, endocrine and neurological diseases[15,16]. Despite this, treatment options remain limited and deeper insights into placental biology are urgently needed.

Placental organoids, first described in 2018, offer a promising model for studying early placental development and the pathogenesis of pregnancy complications[17,18]. While current protocols have advanced our understanding, they often rely on Matrigel, primary tissue from pregnancy terminations or invasive chorionic villous sampling, and costly media components, limiting reproducibility and accessibility[19–29]. Additionally, many trophoblast organoids display an inverted cellular architecture, hindering the study of key physiological processes, including diffusion across placental surfaces and cell secretions.

In this study, we evaluate the organoid-forming potential of the immortalised first-trimester trophoblast cell line, ACH-3P, using both traditional Matrigel embedding and droplet-on-demand bioprinting with a synthetic, polyethylene glycol (PEG) based matrix. Organoids form in both conditions and differentiate into major trophoblast subtypes, with matrix composition influencing lineage specification: Matrigel driving organoids towards syncytialisation and PEG promoting an EVT phenotype.

We further apply our bioprinted organoids to model inflammatory conditions and assess the effects of current and emerging therapies for preeclampsia, aspirin and metformin[30–32]. We report that these treatments at 0.5 mM concentration cannot rescue the impact on organoid metabolic activity or the reduction in STBs. To better mimic placental architecture, we implement a suspension method that reverses organoid polarity, promoting syncytialisation at the outer surface. Finally, we demonstrate that trophoblast stem cells (TSCs) form organoids in both matrices, although growth is restricted in the stiffer PEG matrix using bioprinting.

Our findings underscore the importance of matrix properties in guiding organoid architecture and trophoblast differentiation, offering an automated, precise, scalable and reproducible platform for studying placental development and high-throughput drug testing for placental dysfunction disorders, including preeclampsia and IUGR.

## Results

### ACH-3P cells readily form organoids in Matrigel and bioprinted matrix with basic growth medium

Firstly, we performed a matrix selection to determine which PEG matrix is optimal for ACH-3P organoid formation by comparing three 1.1 kPa PEG formulations: (i) a blank matrix with no adhesion peptides, (ii) a matrix containing α- and β-laminin peptide chains (IKVAV and YIGSR), and (iii) a matrix containing collagen type I peptide mimetic (GFOGER) based on proteins expressed in human placental tissue[19,20,33]. These were manufactured and provided by a company called Inventia Life Sciences in Australia without any further modification; thus, reproducibility is ensured through the manufacturer's QC methods for any lab utilising the RASTRUM bioprinting platform. The matrix components are tailored to the placental microenvironment by matching the stiffness and ECM components of the placental tissue and underlying decidua basalis[21]. A matrix stiffness of 1.1 kPa provides an initial scaffold that supports cell organisation into tissue structures, which are expected to progressively stiffen beyond the parameters defined by the starting matrix. ACH-3P cells formed significantly more organoids (15.7 ± 0.9) in the blank PEG matrix compared with those containing IKVAV and YIGSR (8.6 ± 0.7) or GFOGER (9.5 ± 0.7) peptides (Supplementary Fig. 1a–d; ****$p < 0.001$). No significant difference in cell viability was observed between synthetic matrix conditions (blank 72.1% ± 3.7 vs. IKVAV and YIGSR 70.7% ± 2.5 vs. GFOGER 59.8% ± 5.7, $p = 0.151$; Supplementary Fig. 1d). For this reason, the blank PEG formulation was chosen for the subsequent bioprinting experiments.

ACH-3P cells embedded in Matrigel or bioprinted matrix spontaneously formed organoids within 3–4 days in growth medium used for basic 2D culture (Fig. 1a, Supplementary Movies 1 and 2). Bright-field imaging revealed that morphologically, organoids grown in Matrigel had prominent pseudopodia projecting from their outer surface and individual cells were also seen budding from organoids, both consistent with described features of EVT organoids (Fig. 1a and Supplementary Fig. 1e). In comparison, bioprinted organoids did not appear to project pseudopodia into the matrix, instead they were observed invading through the matrix microenvironment, often fusing upon contact (Supplementary Movie 2). There was no significant difference in 2D area between the Matrigel and bioprinted organoids (31,355 μm² ± 4806 vs. 27012 μm² ± 4152, Fig. 1b, c). Cell viability was assessed using calcein AM (live) and ethidium homodimer-III (dead) staining, revealing a significant increase in the proportion of live cells in Matrigel compared to bioprinted organoids (Fig. 1d, e and Supplementary Fig. 1f; 92.93% ± 1.9 vs. 48.00% ± 7.2, **$p = 0.004$). However, incubation with a resazurin-based solution, Alamar Blue, revealed that both Matrigel and bioprinted organoids were metabolically active (Fig. 1f; 2829 ± 30.7 vs. 2516 ± 149.3, $p = 0.109$). When assessed over time, the metabolic activity and proliferation of cells in both conditions were seen to increase over time, with Matrigel organoids demonstrating a steep incline in cell activity (Supplementary Fig. 1g). Organoid-conditioned medium collected from Matrigel and bioprinted organoids tested positive for β-human chorionic gonadotropin (β-hCG) using over-the-counter pregnancy tests (Fig. 1g).

### ACH-3P cells spontaneously differentiate to major trophoblast subtypes in Matrigel and bioprinted organoids

Organoids were investigated for the presence of trophoblast subtypes by whole-organoid immunofluorescence and confocal microscopy. E-cadherin, a marker of cytotrophoblasts (CTBs), was present extensively throughout the organoids grown in Matrigel and bioprinted conditions (Fig. 1h). EVTs were detected on the periphery of both Matrigel and bioprinted organoids with no significant difference in the number of human leukocyte antigen G (HLA-G)+ trophoblasts between each condition (Fig. 1h, i; 12.33% ± 0.8 vs. 11.23% ± 3.7, $p = 0.784$). Single β-hCG+ cells and syncytialised clusters were seen sporadically throughout the organoids, representing the STB subtype (Supplementary Movie 3). Image analysis of DAPI-labelled organoids revealed no significant difference in the number of β-hCG+ nuclei between Matrigel and bioprinted organoids (Fig. 1h, i; 10.37% ± 2.6 vs. 9.23% ± 2.1, $p = 0.753$).

Organoid staining with haematoxylin and eosin (H&E) allowed further visualisation of the cellular organisation and architecture of organoids, and processing of whole matrix plugs provided the opportunity to view organoids in situ. Organoid sectioning revealed cavities in the inner cores of organoids (Fig. 1j and Supplementary Fig. 1h). While cavities were seen in both groups, the bioprinted organoids generally appeared more compact and contained fewer cavities. Areas of syncytialisation were observed in organoids from both groups (Fig. 1j).

### Transcriptomic profiling of organoids provides further insight into trophoblast subtypes and differentiation trajectories

Organoid transcriptomes were profiled to screen for trophoblast subtypes within Matrigel and bioprinted organoids at the single-cell

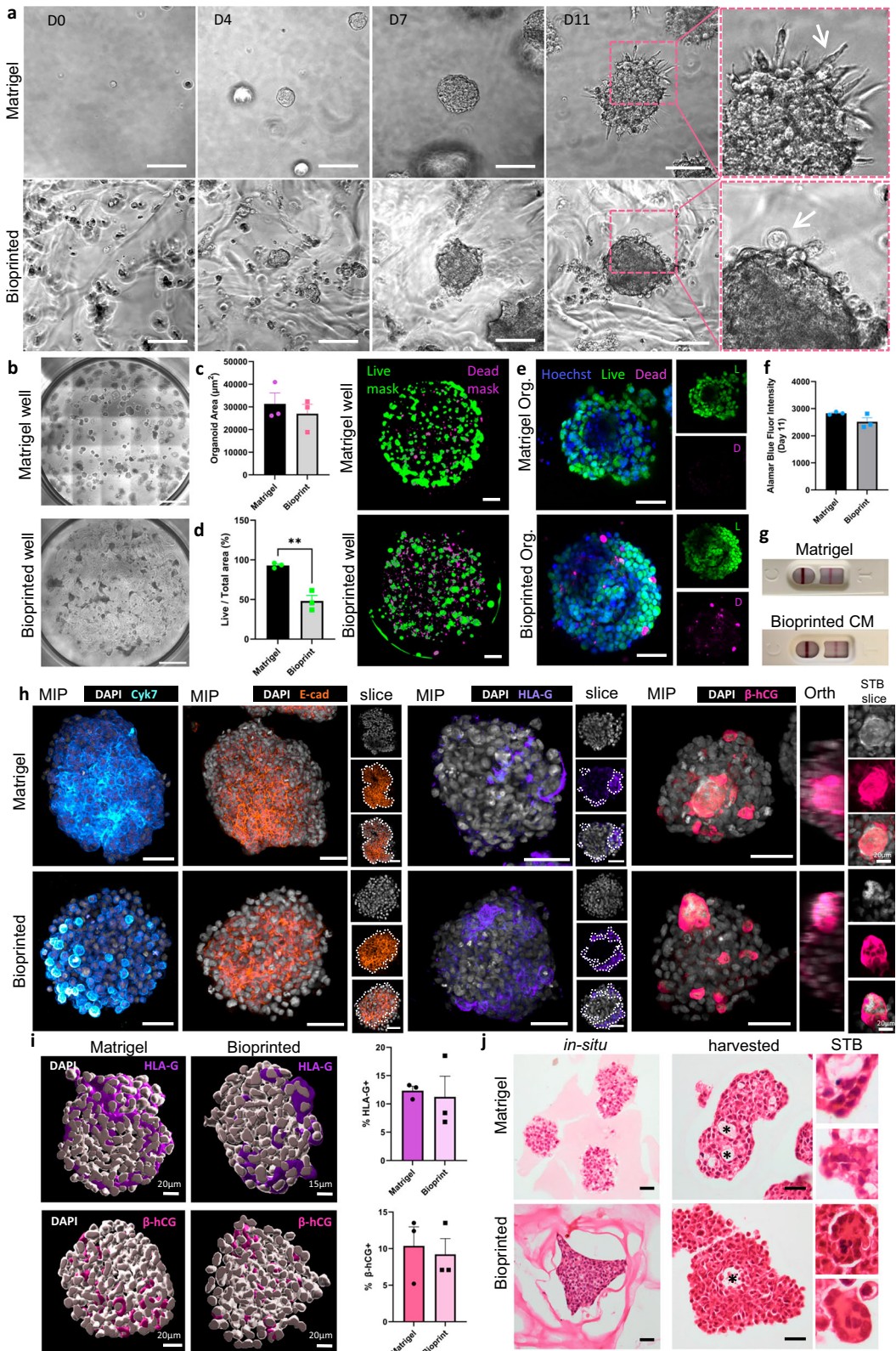

level using the 10x Genomics Chromium platform. After quality control and normalisation, a total of 6081 cells and 17,159 genes were retained for subsequent analysis of Matrigel and bioprinted organoid samples. As previously established by the creators of the ACH-3P cell line[22], these cells are devoid of vimentin, stably express Cyk7 and contain an HLA-G-expressing population of EVTs, which is consistent with our results (Fig. 2 and Supplementary Fig. 2a). Uniform manifold

approximation and projection (UMAP) analysis of Matrigel and bioprinted organoids was performed to further elucidate the populations of trophoblast subtypes. Through manual annotation based on published markers, the three major trophoblast cell types, including CTB, STB, and EVT, were identified (Fig. 2a): the proportion of CTB cells in each sample remained consistent, at 44% in Matrigel and 43% in bioprinted organoids. Although the EVT population made up 33% of cells

**Fig. 1 | ACH-3P spontaneously differentiate within Matrigel and bioprinted organoids. a** Single cells embedded in Matrigel or synthetic matrix (bioprinted) formed organoids and grew over 12 days in culture across three independent replicates. Outer cell projections and surfaces are denoted by white arrows in the inset image; scale bar = 100 μm. **b** Montage of whole-well; scale bar = 1 mm. **c** Organoid size measured at day 11 and plotted as organoid area. Unpaired, two-tailed *t*-test, data presented as mean ± SEM, individual data points represent means from three experimental repeats. **d** Whole-well images acquired and analysed using an IncuCyte imaging system; scale bar = 800 μm. Area occupied by green fluorescence (live cells) was compared to the total area occupied by cells to determine viability (%). Data presented as mean ± SEM, individual data points represent means from three experimental repeats, unpaired two-tailed *t*-test, **$p < 0.01$ ($p = 0.0038$). **e** Organoids labelled with markers for live (green) and dead cells (magenta); scale bar = 100 μm. **f** Day 11, Alamar blue fluorescence intensity. Unpaired two-tailed *t*-test, $n = 3$ experimental repeats, data presented as mean ± SEM. **g** Over-the-counter pregnancy tests incubated with organoid conditioned medium (CM). **h** Organoids immunolabelled for DAPI (grey), cytokeratin 7 (Cyk7, cyan hot), E-cadherin (E-cad, orange), human leucocyte antigen G (HLA-G, purple) and beta human chorionic gonadotropin (β-hCG, pink) were assessed from three independent repeats. Images further processed using the NIS Elements denoise.ai algorithm. Maximum intensity projections of 60 μm z stacks alongside a single optical section. Scale bar = 50 μm. A YZ orthogonal view and XY single optical section of syncytiotrophoblasts (STB) in β-hCG labelled organoids. **i** Three-dimensional visualisation of z-stacks using Bitplane Imaris rendering. Number of nuclei compared to number of cells positive for either HLA-G or β-hCG in each setting. Unpaired, two-tailed *t*-test, $n = 3$ experimental repeats containing at least 10 whole-mounted organoids each, data presented as mean ± SEM. **j** Histologically sectioned and processed organoids stained by haematoxylin and eosin (H&E). Internal cavities denoted by asterisks; scale bar = 50 μm. D Day, Org. organoid. Source data are provided as a Source Data file.

in Matrigel organoids, they contributed a higher proportion at 54% in bioprinted organoids. In addition, the STB population formed 22% of cells in Matrigel organoids but only 3% of bioprinted.

In further investigations, we observed two additional subclusters —a metal regulating CTB and maturing EVT population—totalling five distinct cell clusters (Fig. 2b). Single cell ordering by pseudotime trajectory analysis was performed to track the relatedness and differentiation of trophoblast subtypes, using SPINT1 expression as the origin point and root of cell differentiation (Fig. 2c)[23]. A differentiation pathway emerged from the major CTB cluster down two main arms: one extending to the EVT subtypes and another extending to the STB cluster.

The CTB clusters were characterised by their increased expression of paternally expressed gene 10 (*PEG10*), integrin alpha-6/beta-1 (*ITGA6*) and cadherin 1 (*CDH1*) genes (Fig. 2b, d and e). A metal-regulating CTB subcluster (CTB_2) was distinguished by increased expression of metallothionein 2A (*MT2A*), a metal transporter protein (Fig. 2b and f).

The EVT clusters progressively displayed less *PEG10*, *ITGA6* and *CDH1*, with increasing expression of EVT markers, integrin subunit alpha 5 (*ITGA5*), *HLA-G* and matrix metallopeptidase 2 (*MMP2*) (Fig. 2d and g). One EVT cluster (EVT_1) displayed increased expression of proliferation marker, ornithine decarboxylase 1 (*ODC1*), and retained a high G2M phase profile (Fig. 2h and Supplementary Fig. 2b, c). This suggests it may represent a progenitor population of cells transitioning from CTB to maturing EVTs. The second EVT subcluster (EVT_2) displayed increased expression of Epstein-Barr virus-induced gene 3 (*EBI3*), an interleukin 27 subunit shown to have roles in maternal immune tolerance during pregnancy (Fig. 2i)[24–27]. The EVT_2 subcluster also expressed increasing levels of integrin subunit alpha-1 (*ITGA1*), a marker of mature EVTs (Fig. 2d).

On the other hand, the STB cluster showed increased gene expression of STB markers including chorionic gonadotropin subunit beta 3 (*CGβ3*), endogenous retrovirus group FRD member 1 (*ERVFRD-1*), *SDC1*, growth differentiation factor 15 (*GDF15*) and matrix metallopeptidase 15 (*MMP15*) (Fig. 2d, j, k). Differentially expressed gene (DEG) analysis revealed expression differences in the main cell clusters depending on their organoid type of origin (Supplementary Fig. 2d).

To determine how confidently our Matrigel- and bioprint-derived trophoblast organoids recapitulate in vivo trophoblast biology, we compared the single-cell transcriptomes of CTB, EVT and STB in our organoids with (i) two primary placental tissue transcriptomes—the first-trimester chorionic villus (8–12-week) dataset[28] and an independent 6-week placenta dataset[29], and (ii) a published patient-derived trophoblast organoid scRNA-seq dataset[34]. MetaNeighbor analysis[35], which ranks cross-dataset cell-type relatedness by the area under the receiver-operating-characteristic curve (AUROC), revealed strong concordance across all comparisons. Using an a priori threshold of AUROC > 0.88 to denote high similarity, each of CTB, EVT, and STB from our organoids formed exclusive, high-confidence links with their cognate in vivo counterparts in every reference dataset, while non-matched pairings remained below background (<0.5) (Fig. 2l and Supplementary Fig. 2e–g). Collectively, the genetic comparisons to in vivo trophoblasts demonstrate that our ACH-3P organoid system preserves the key transcriptional identities of primary placental trophoblast lineages. This finding supports its utility as an experimental placental tissue platform that closely recapitulates the human placenta and can be used to elucidate mechanisms of placentation and as a drug screening platform for placental dysfunction disorders such as preeclampsia.

## Proteomic profiling of organoids

Following a screening of trophoblast transcriptomes, we further analysed the expression profiles of Matrigel and bioprinted organoids at the protein level. Matrigel or bioprinted organoids were isolated from their matrices and characterised alongside 2D cultured ACH-3P cells and Matrigel matrix using liquid chromatography with tandem mass spectrometry. A total of 4984 proteins were detected across all samples and z-scores mapped (Fig. 3a). There were only 33 proteins significantly upregulated in Matrigel organoids compared to bioprinted, while 1.238 proteins were significantly upregulated in bioprinted compared to Matrigel organoids (Fig. 3b and c). When searched in Reactome (https://reactome.org/), this corresponded to a significant upregulation of 2 pathways in Matrigel organoids, namely the biosynthesis of ubiquinol and haem. On the other hand, 114 biological pathways ($p < 0.05$) were upregulated in bioprinted organoids, with many relating to mRNA processing.

Though a distinct TSC population was not noted at the transcriptome level, proteomic analysis revealed a significant increase in TSC markers such as ubiquitin-like interferon-stimulated gene 15 (ISG15), stearoyl-CoA desaturase (SCD) and sorting nexin-2 (SNX2) in bioprinted compared to Matrigel organoids (Fig. 3d). There was also a significant increase in abundance of CTB marker, CDH1, and proliferation marker, Ki67, in bioprinted organoids. Consistent with the scRNAseq data, bioprinted organoids had a greater abundance of EVT proteins, including HLA-G, ITGA5 and quiescin sulfhydryl oxidase 1 (QSOX1). Interestingly, there were no significant differences observed in the protein abundance of STB markers. Syncytin 2 (SYCY2), 3 beta-and steroid delta-isomerase 1 (3BHS1) or cytochrome P450 family 19 subfamily A member 1 (CP19A), between Matrigel and bioprinted organoids. In fact, the Matrigel expressions of these markers were highly variable between repeats.

## Trophoblast organoid treatment with aspirin and metformin in an inflammatory environment

To establish a preeclampsia-like organoid model, bioprinted organoids were initially screened against varying concentrations of TNFα (1, 5, 20 ng/ml), an inflammatory cytokine that is highly expressed in

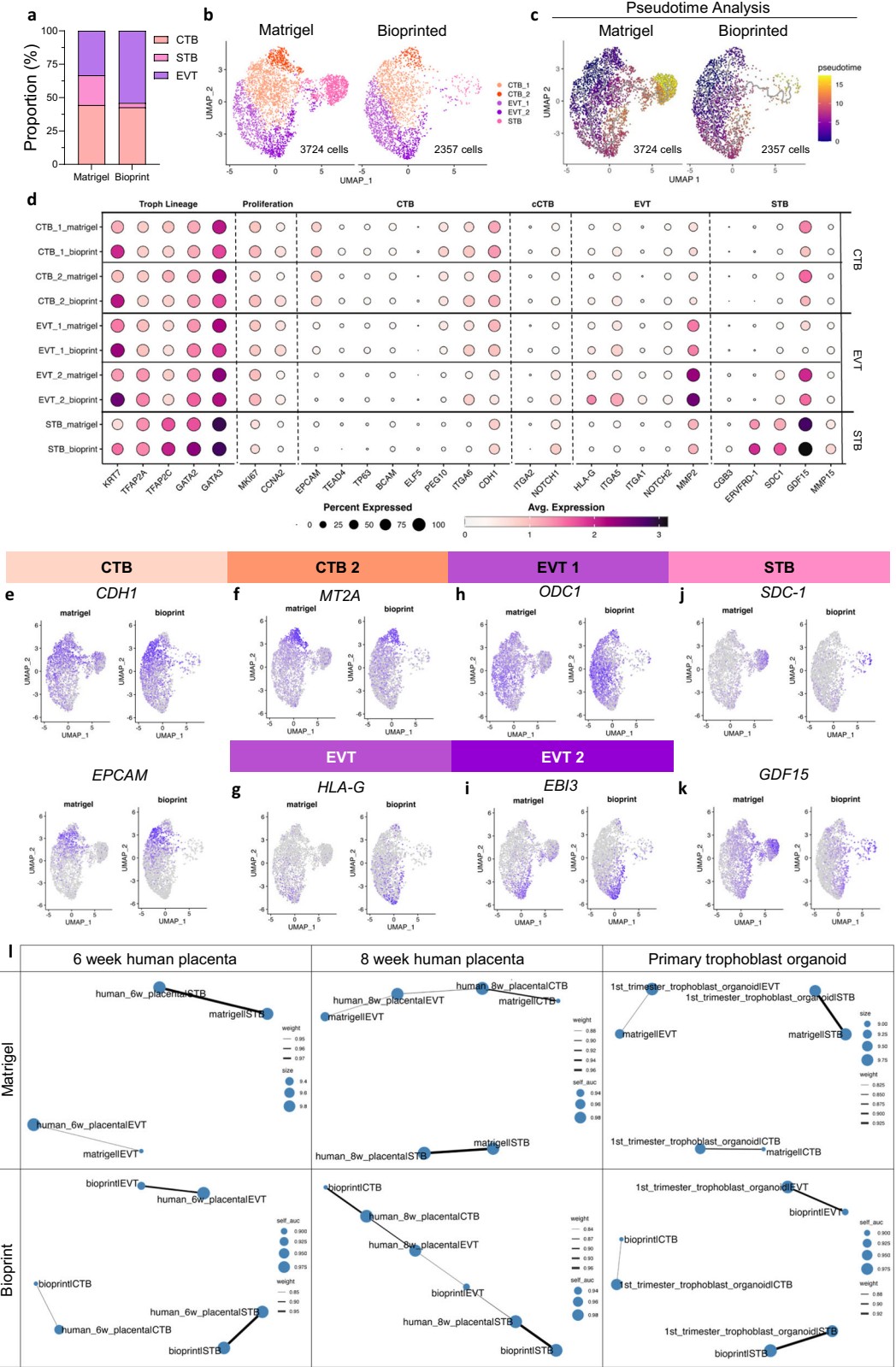

women with preeclampsia (Supplementary Fig. 3)[30]. While there was no significant difference in the number ($p = 0.520$), size ($p = 0.244$) or organoid viability ($p = 0.890$) between control or TNFα (1, 5, 20 ng/ml) groups (Supplementary Fig. 3a–c), and all concentrations of TNF-α reduced organoid proliferation ($p < 0.05$; Supplementary Fig. 3d); the concentration of C reactive protein (CRP) secreted by organoids exposed to 20 ng/ml TNFα was around 25× higher than all other

groups (Supplementary Fig. 3e), demonstrating an inflammatory status. For this reason, we proceeded with 20 ng/ml TNFα. Aspirin is prescribed prophylactically to women at high risk of developing preeclampsia due to its established effects in preventing preterm preeclampsia (delivery <37 weeks)[31]. On the other hand, therapeutics already deemed safe for pregnancy, such as metformin, are being assessed for their repurposing capacity to treat severe preeclampsia

**Fig. 2 | Single-cell transcriptomic profiles of Matrigel and bioprinted ACH-3P organoids. a** Proportions of main trophoblast subtypes (CTB - cytotrophoblast, STB - syncytiotrophoblast, and EVT - extravillous trophoblasts) detected within Matrigel and bioprinted organoids. **b** Uniform manifold approximation and projection (UMAP) depicting an additional subcluster of CTBs and maturing EVTs. **c** UMAP plots of differentiation trajectory in organoids, depicted in pseudotime scale. **d** Dot plot representation of trophoblast lineage and subtype gene expression across the five distinguishable clusters. **e** UMAP plots of CTB markers, E-cadherin (*CDH1*) and epithelial cell adhesion molecule (*EPCAM*). **f** UMAP plot of metallothionein 2A (*MT2A*). **g** UMAP plot of human leucocyte antigen G (*HLA-G*). **h** UMAP plot of ornithine decarboxylase 1 (*ODC1*). **i** UMAP plot of Epstein-Barr virus-induced gene 3 (*EBI3*). **j** and **k** UMAP plots of STB markers, syndecan 1 (*SDC-1*) and growth differentiation factor 15 (*GDF15*). Single-cell RNA sequencing data are deposited in the GEO under accession number GSE279994. **l** Area under the receiver-operating-characteristic curve (AUROC) similarity networks between cell types from Matrigel or bioprinted organoid transcriptomic datasets compared to published data from human placenta or primary trophoblast organoids. Priori threshold AUROC > 0.88. Source data are provided as a Source Data file.

with promising results and can ameliorate stress-induced placental cell dysfunction[32,36,37].

Bioprinted organoids were used for a drug screening with aspirin (0.5 mM) or metformin (0.5 mM) to investigate their ability to ameliorate inflammation-induced (TNFα 20 ng/ml) trophoblast dysfunction in the context of preeclampsia (Fig. 4a). Although there was a statistically significant difference across all the groups in the organoid number (Fig. 4b; $p = 0.043$), post-hoc tests showed no differences between any groups. Organoid size remained similar across different organoid conditions (Fig. 4c; $p = 0.109$). Trophoblast organoid metabolic activity or proliferation was reduced from Day 9 onwards following TNF-α or aspirin/metformin treatment (Fig. 4d, e; $p < 0.01$), which was not rescued by aspirin ($p < 0.001$) or metformin ($p < 0.01$) and aligned with our previous findings in 2D culture[36]. Furthermore, we also determine the impact TNF-α ± aspirin/metformin treatment has on inflammatory markers, CRP and interleukin 6 (*Il-6*). CRP was significantly higher in organoid media containing TNF-α compared to all other conditions, which was substantially reduced in the presence of aspirin/metformin possibly suggesting that aspirin/metformin can reduce inflammation associated with trophoblast organoids (Fig. 4f). Although there were no statistically significant differences across the groups in *Il-6* gene expression within trophoblast organoids, *Il-6* gene expression is overall lower in the presence of TNF-α + aspirin, compared to TNF-α alone, which was not applicable to metformin (Fig. 4g). Next, we investigated the impact TNF-α ± aspirin/metformin had on ACH-3P cytotrophoblast differentiation within organoid culture (Fig. 4h–j). Interestingly, the number of β-hCG⁺ STBs was reduced in all groups ($p < 0.05$) except TNF-α + metformin ($p = 0.07$), compared to control (Fig. 4h, j), suggesting the presence of STB stress induced by TNF-α, which was not restored by aspirin/metformin at 0.5 mM concentration. On the other hand, no changes in the number of HLA-G⁺ EVTs were observed following treatment with aspirin/metformin and/or TNF-α (Fig. 4i, j).

### Organoid suspension reverses cellular organisation
To reverse the polarity and organisation of trophoblast subtypes within organoids, we harvested 3-day-old organoids from their respective conditions and cultured them in suspension for up to 12 or 28 days. In the absence of a matrix to confine them, suspended organoids grew comparably larger over time and were occasionally seen fusing together (Fig. 5a). Further to this, since there was no adhesive matrix present to interact with, the organoids retrieved from both gel conditions displayed spherical morphology. Histological processing also revealed that internal cavities continued to increase in size as organoids grew (Fig. 5b).

To examine the inversion of cellular organisation within the suspended organoids, another marker of STBs, syndecan-1 (SDC-1), was immunolabelled and imaged in whole organoids. Individual z slices of representative organoids are presented in Fig. 5c. SDC-1 expressing cells were observed on the periphery of organoids grown in suspension, with areas of syncytialisation depicted by arrows. Higher resolution images reveal the potential presence of a microvillus brush border on the outer edge of these syncytialised regions (Supplementary Fig. 4a). Suspended organoids were also immunolabelled for E-cadherin, HLA-G and β-hCG to confirm the presence of each trophoblast subtype (Supplementary Fig. 4b).

### Trophoblast stem cell-derived organoids are growth-restricted in a stiff matrix
To assess the translation of our bioprinting conditions to TSC organoid formation, we manually embedded CT29 'Okae' TSC line in Matrigel or bioprinted cells in the same blank 1.1 kPa matrix. Organoids formed in both conditions, but they were significantly smaller in the bioprinted matrix, even after stimulation with STB or EVT differentiation medium (STB-DM and EVT-DM, respectively; Fig. 6a, b). Organoids from both conditions secreted β-hCG (Fig. 6c). There was also a significant reduction in organoid formation efficiency in bioprinted conditions (Fig. 6d). Both Matrigel-embedded and bioprinted organoids could differentiate into SDC-1⁺ STB organoids, and HLA-G⁺ EVT organoids under predefined medium (Fig. 6e–g). There was no difference in the number and size between undifferentiated trophoblast organoid medium (TOM), STB-DM or EVT-DM treated organoids within each condition (Fig. 6e, f). However, within bioprinting conditions, 'rich' matrix containing fibronectin, laminin, collagen IV peptides, laminin-511 and hyaluronic acid increased the number ($p < 0.0001$) and size ($p < 0.05$–0.001) of TSC-derived organoids compared to either subtype alone bioprinted with 'blank' PEG matrix (Fig. 6f).

## Discussion
Early placental development is a complex and tightly regulated process that can be disrupted in various pregnancy complications. 3D cultures of cells from the maternal-foetal interface provide tools to understand the elusive mechanisms dictating placental physiology more thoroughly and screen potential treatments[38]. In this study, we describe the generation of trophoblast organoids from the ACH-3P first-trimester trophoblast cell line and TSCs within Matrigel or bioprinted into a synthetic, PEG-based matrix. Bioprinting offers an innovative opportunity to produce low-cost, high-throughput and reproducible organoid models with high precision and resolution. Here, we aimed to demonstrate the feasibility of bioprinting trophoblast organoids and assess the impact of ECM on trophoblast differentiation.

Establishing the most appropriate matrix environment for 3D placental models is crucial, given the growing recognition of mechanical cues and cell–matrix interactions influencing stem cell differentiation, trophoblast fusion and trophoblast spheroid formation[39]. Alternative natural and synthetic hydrogels, such as PEG-based formulations, have been gaining attention for their tuneable composition and stiffness properties[40]. Based on our initial matrix selection, we demonstrated increased ACH-3P organoid formation efficiency within a naked PEG matrix containing no adhesion peptides or proteins. This condition may be due to the production of ECM components by trophoblasts, both degrading and contributing to the ECM environment they embed and invade into[41]. In fact, the decidua basalis, where trophoblasts embed within the uterus, has a reported stiffness of 1.25 kPa, likely due to the trophoblast deposition of ECM components including laminin, fibronectin, collagen IV, heparan sulfate, and fibrillin 1[21].

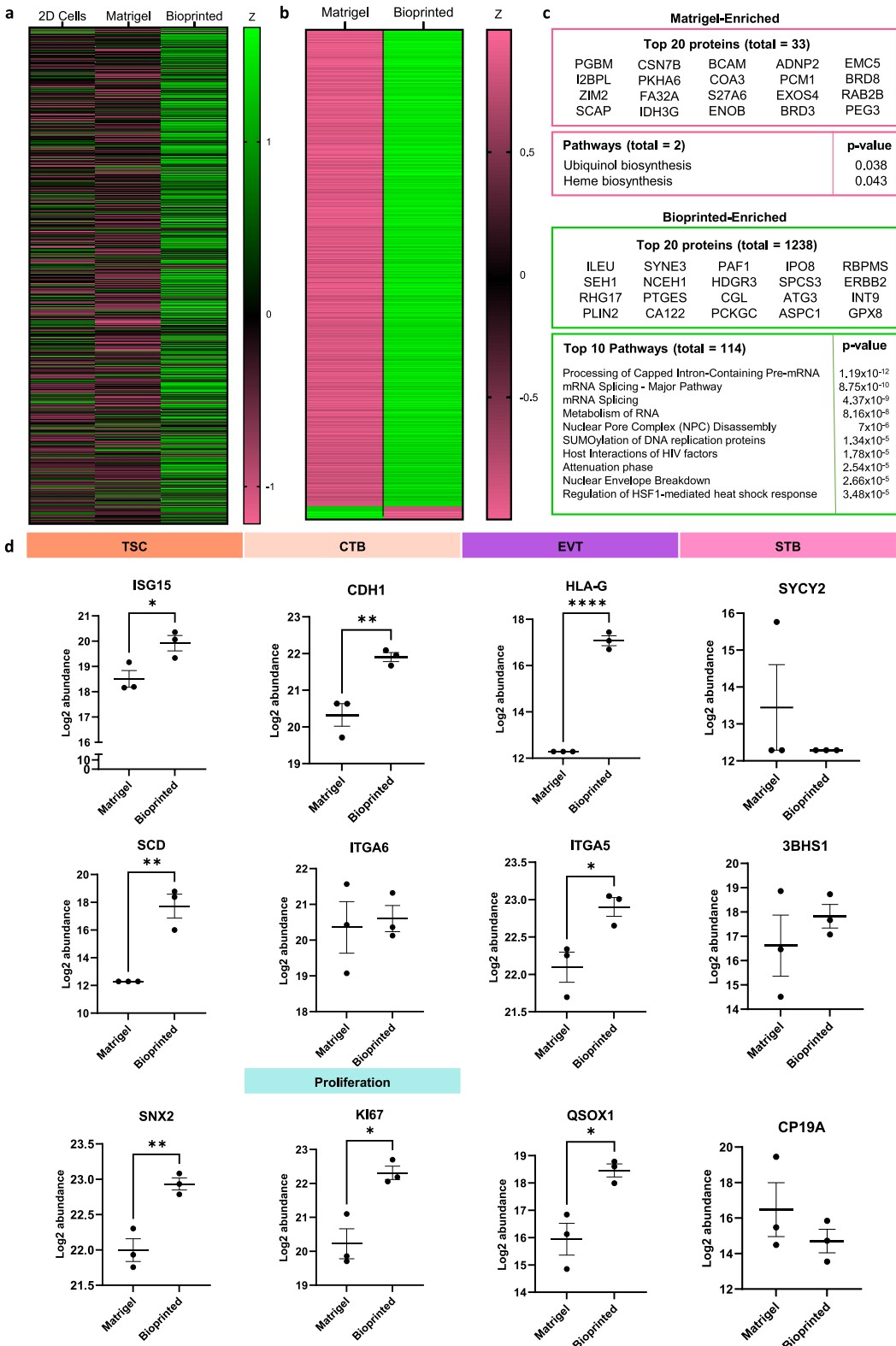

**Fig. 3 | Proteomic profiling of Matrigel and bioprinted ACH-3P organoids.**
Organoids were digested and analysed by liquid chromatography with tandem mass spectrometry. **a** Heat map of mean protein z-score for 2D cells (2D), Matrigel organoids and bioprinted organoids. **b** Heat map of mean z-score for significantly differentially expressed proteins between Matrigel and bioprinted organoids. **c** Top proteins and pathways when ranked by significance for each group as determined by an unpaired, two-tailed Student's $t$-test. **d** Differences in key trophoblast subtype marker expression between Matrigel and bioprinted organoid groups. Unpaired, two-tailed Student's $t$-test, $n = 3$ experimental repeats, each containing 800 pooled organoids, $*p < 0.05$, $**p < 0.01$, $***p < 0.001$, $****p < 0.0001$, data plotted as experimental mean ± SEM. (ISG15 $p = 0.0344$; SCD $p = 0.0033$; SNX2 $p = 0.0067$; CDH1 $p = 0.0089$; Ki67 $p = 0.0124$; HLA-G $p < 0.0001$; ITGA5 $p = 0.0274$; QSOX1 $p = 0.0161$). All protein mass spectrometry raw data is available in a ProteomeXchange partner repository with identifier PXD056796.

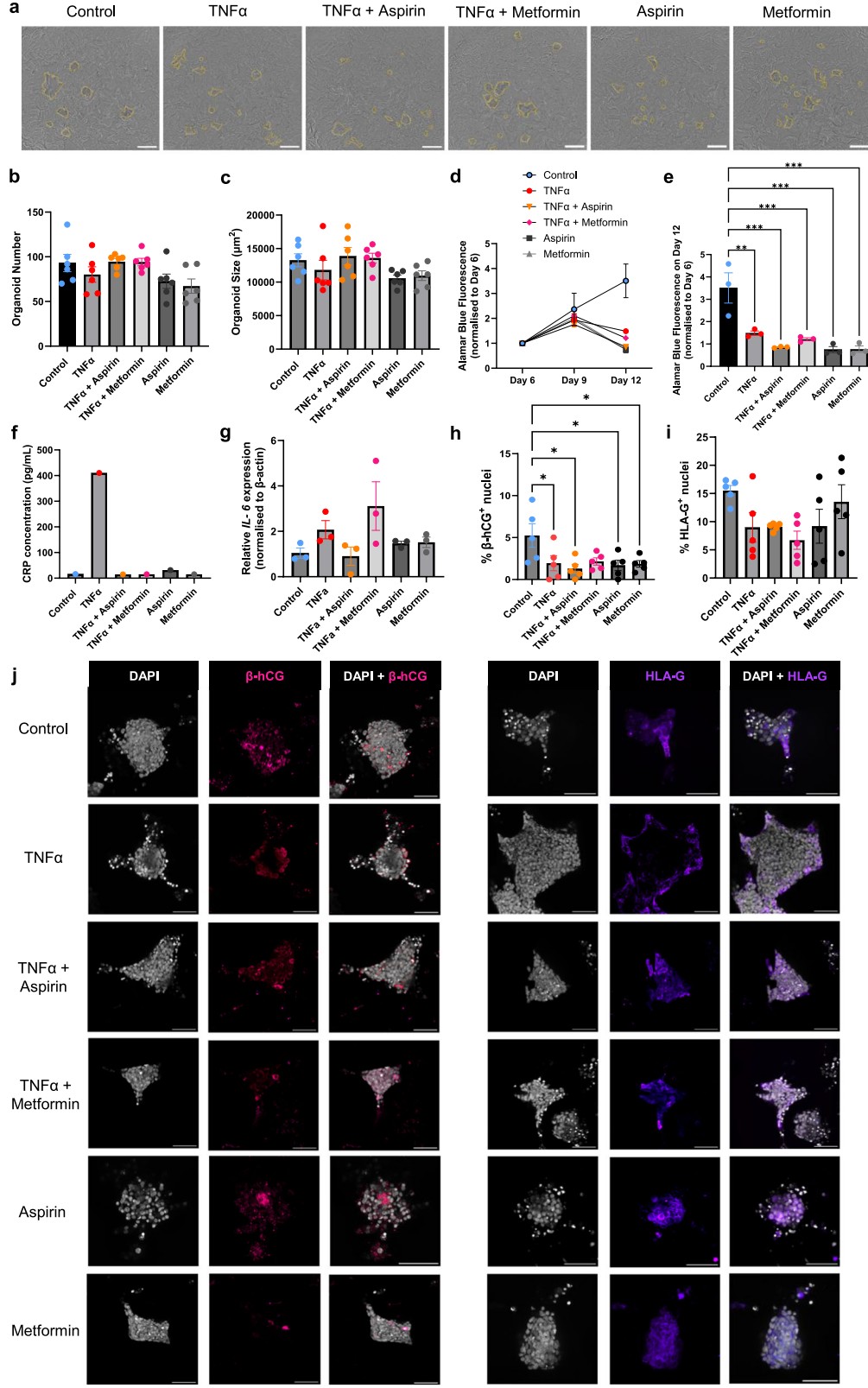

Within both Matrigel and bioprinted PEG and basic growth medium, ACH-3P single cells gave rise to organoids that grew steadily over time with no observable differences in size. Both Matrigel and bioprinted organoids contained the three major trophoblast subtypes of CTBs, EVTs and STBs as shown by immunofluorescence labelling. This is consistent with the results of recent work comparing the organoid formation of ACH-3Ps alongside HTR-8/SVneo and Swan-71

trophoblast cell lines in Matrigel[42]. Despite rapid organoid formation, bioprinted cultures encountered reduced viability, likely due to shear stress applied to the cells during bioprinting, an important consideration for primary cell work. Alternatively, this result may highlight a need to further elucidate the ECM composition and stiffness requirements at different stages of organoid expansion and differentiation, as investigated in intestinal organoids[43]. Transcriptomic

**Fig. 4 | Bioprinted organoid formation after treatment with TNFα, aspirin and metformin. a** Brightfield images of bioprinted organoids incubated with inflammatory cytokine tumour necrosis factor alpha (TNFα, 20 ng/ml) ± aspirin (0.5 mM) or metformin (0.5 mM). Images acquired using an IncuCyte. Organoids annotated with an orange outline. Scalebar = 500 μm. **b** Mean number of organoids counted across six bioprint units. Ordinary one-way ANOVA with Šídák's multiple comparisons post-hoc test, data presented as mean ± SEM, individual data points represent mean per bioprint unit. **c** Mean 2D organoid area. Ordinary one-way ANOVA with Šídák's multiple comparisons post-hoc test, data presented as mean ± SEM, individual data points represent mean per bioprint unit. **d** Alamar Blue fluorescence measured at 590 nm, normalised to the day 6 fluorescence intensity of each condition. Data presented as mean ± SEM, individual data points represent mean across three bioprint units. **e** Alamar Blue fluorescence on Day 12 normalised to the day 6 fluorescence intensity of each condition. Ordinary one-way ANOVA with Šídák's multiple comparisons post-hoc test, data presented as mean ± SEM, individual data points represent mean per bioprint unit, **p < 0.01, ***p < 0.001 (Control vs. TNFα p = 0.0026; Control vs. TNFα + Aspirin p = 0.0002; Control vs. TNFα + Metformin p = 0.0009; Control vs. Aspirin p = 0.0002; Control vs. Metformin p = 0.0002).

**f** C-reactive protein (CRP) concentration of condition medium from 3 pooled bioprint units per condition. Data plotted as individual data point from pooled sample. **g** Relative interleukin 6 (*Il-6*) mRNA expression of organoids harvested from each bioprint unit. Data presented as mean ± SEM, individual data points represent the mean per bioprint unit. Ordinary one-way ANOVA with Šídák's multiple comparisons post-hoc test. **h** Number of β-hCG positive nuclei divided by total number of nuclei per organoid. Ordinary one-way ANOVA with Šídák's multiple comparisons post-hoc test, data presented as mean ± SEM, data points represent individual organoids from pooled bioprint units, p*<0.05 (Control vs. TNFα p = 0.0496; Control vs. TNFα + Aspirin p = 0.012; Control vs. Aspirin p = 0.0281; Control vs. Metformin p = 0.0417). **i** Number of HLA-G positive nuclei divided by total number of nuclei per organoid. Ordinary one-way ANOVA with Šídák's multiple comparisons post-hoc test, data presented as mean ± SEM, data points represent individual organoids from pooled bioprint units. **j** Immunofluorescence images labelled with DAPI (grey), β-hCG (pink) and HLA-G (purple) acquired using a Leica Stellaris confocal microscope at ×25, scale bar = 100 μm. Source data are provided as a Source Data file.

analysis confirmed the spontaneous differentiation of major trophoblast subtypes, originating from a CTB signature and directed towards either EVT or STB transcriptomic profiles. Interestingly, detailed clustering analysis revealed a distinct subpopulation of CTB in both organoids, distinguishable by the expression of *MT2A*, a metallothionein. The *MT2A* form of metallothionein is strongly expressed in the placenta and increases throughout gestation[44,45]. *MT2A* in the placenta is believed to have roles in detoxifying trace elements, protecting the developing foetus from heavy metals, clearing reactive oxygen species, trophoblast differentiation and as a regulator of labour-associated genes[46–50]. Metallothionein, and specifically *MT2A*, expression has previously been reported in HTR-8/SVneo, BeWo and JAR trophoblast cell lines and induced human pluripotent stem cells resembling trophoblasts[45,47,51,52]. Further, when Okae and colleagues were establishing their TSC lines, they performed transcriptomic comparisons of isolated primary trophoblast subtypes versus cultured TSCs stimulated to differentiate. RNAseq-based profiling revealed that *MT2A* had the highest expression in primary CTBs compared to other trophoblast subtypes, and in 3D TSC cultures being directed to differentiate into STBs[53].

Our data showed that single cells emerged from organoid structures, thereby reflecting the physiology of EVTs in vivo. Immunofluorescence labelling for HLA-G confirmed the presence of an EVT population of cells in both types of organoids, but no statistically significant difference in the proportion of these cells was observed by imaging (12.3% Matrigel vs. 11.2% bioprinted). However, single-cell transcriptomics indicated that bioprinted organoids contained a higher proportion of EVT cells (33% Matrigel vs. 54% bioprinted). This analysis also provided a greater insight into trophoblast subtypes, detecting at least two distinct populations of EVTs with varying expression of *HLA-G* that may have been indistinguishable by confocal fluorescence microscopy alone. Given this subcluster demonstrated greater expression of *ITGA1*, a marker of EVT maturity, we consider EVT_2 a more mature EVT subcluster[22,54–56]. The impact of a stiffer PEG matrix in directing EVT differentiation is corroborated by a recent study showing that trophoblast migration and invasion are increased in correlation with an increasing ECM stiffness[57]. Greater EVT presence was reinforced at the protein level using proteomics analysis, with bioprinted organoids showing increased abundance of several EVT markers.

Concomitant to this, the activity of STBs in the organoids was first signalled by the positive result of a pregnancy test. At the morphological level, Matrigel and bioprinted organoids contained internal cavities, although bioprinted organoid cavity formation appeared less frequent. These cavities are consistent with those observed in published trophoblast organoid models and resemble the lacunae formed by early STBs where maternal blood pools for diffusion across the syncytial membrane[17,18,58]. Immunofluorescent quantification of β-hCG⁺ nuclei revealed no significant difference between Matrigel or bioprinted organoids (Matrigel 10.4% vs. bioprinted 9.2%), which is aligned with in vivo STB proportions of 9.7%[29]. This was supported by our proteomic analysis demonstrating no significant differences in STB protein markers between Matrigel and bioprinted organoids, although the key STB marker, hCG, was not detected. Contrarily, scRNAseq detected significantly fewer STB cells in bioprinted organoids compared to those in Matrigel. However, a caveat of the application of scRNAseq to placental tissue is that it is unable to separate individual nuclei within a shared cytoplasm, as occurs in multinucleated STBs. This limitation makes transcriptomic conclusions around trophoblast subtypes present in our organoids difficult to draw. Thus, methods such as single-nucleus RNA sequencing (snRNAseq), recently applied to STB analyses and to other multinucleated skeletal muscle myofibers, may be more suitable[28,59].

Comparison of our scRNAseq datasets to publicly available in vivo data from 6- and 8-week human first-trimester placental tissue and first-trimester primary trophoblast organoids demonstrated strong similarity to these tissues. In fact, the bioprinted ACH-3P organoids showed stronger associations with the 6-week placental tissue than our Matrigel-derived organoids, with the CTB subtype lacking significant similarity in Matrigel organoids compared to those from this tissue. The utility of our bioprinted trophoblast organoids to be used for drug screening of conditions caused by placental dysfunction, such as preeclampsia, was also demonstrated. We applied inflammatory stimuli, TNF-α, to mimic a preeclamptic placenta and treated organoids with clinically used treatments, aspirin and metformin. We showed that TNF-α induced a pro-inflammatory environment that impeded metabolic activity or proliferation of bioprinted trophoblast organoids, and reduced the percentage of STBs, which aspirin or metformin were unable to rescue at the concentration of 0.5 mM. STB stress plays a central role in the pathogenesis of preeclampsia, suggesting that our model of TNF-α-induced stress in bioprinted trophoblast organoids resembles a preeclamptic placenta[60,61]. Our previous work in 2D in vitro ACH-3P models shows the ability of aspirin and metformin to rescue stress-induced trophoblast dysfunction; however, not in the presence of TNF-α but rather hypoxia mimic or oxidative stress[36]. Aspirin has been shown to increase EMT transition and trophoblast invasion, likely causing a shift in proliferation towards EVT formation in these organoids[62]. Here, aspirin in the presence of TNF-α showed a reduction in the number of STBs, but it did not affect the EVTs abundance. Metformin has been shown to regulate trophoblast metabolism and increase expression of STB markers[63]. In our drug screen, metformin did not have a significant impact on the number of STBs or EVTs

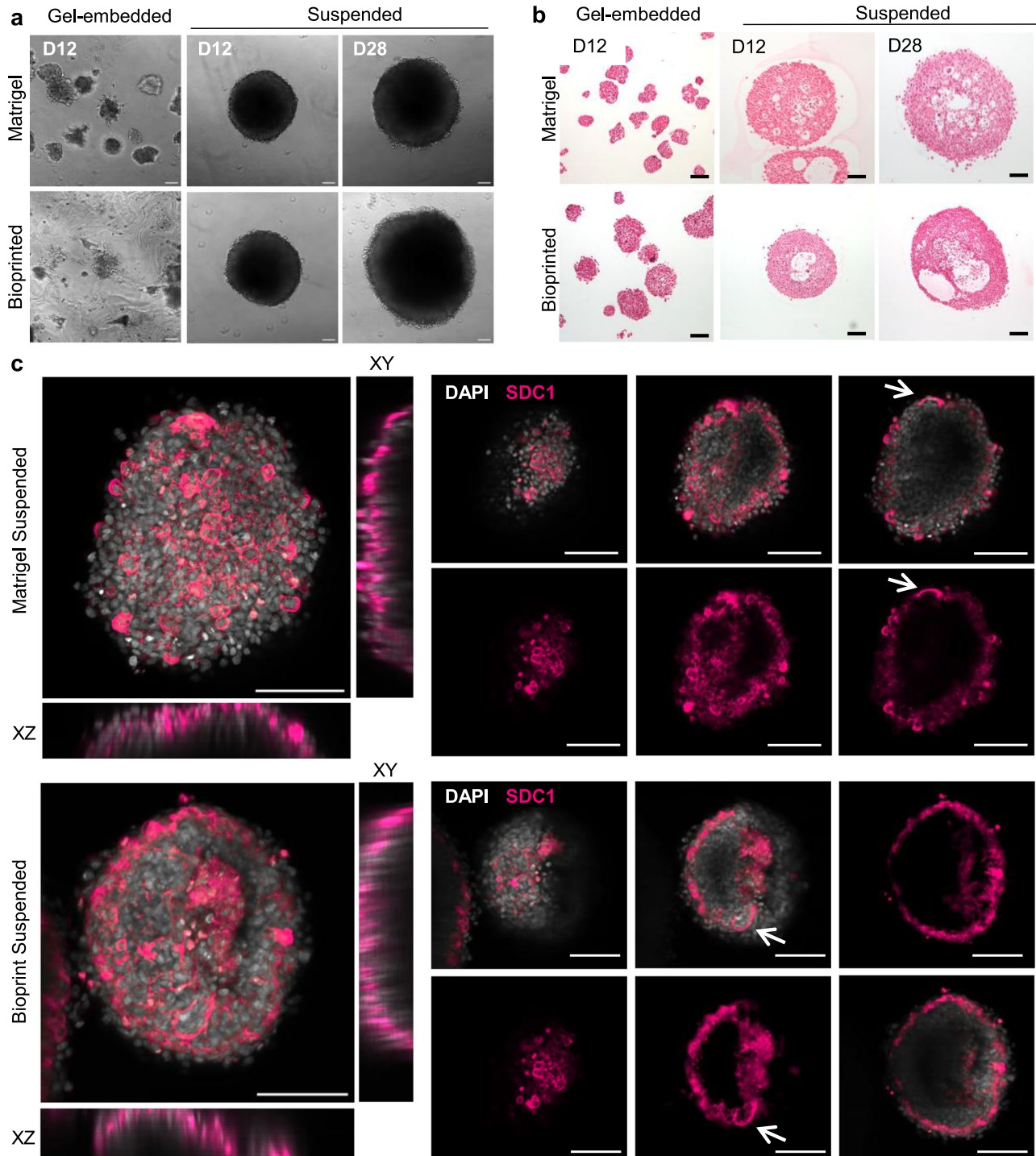

**Fig. 5 | Syncytiotrophoblasts form on the outer surface of trophoblast organoids in suspension culture.** Organoids grown in Matrigel or bioprinted conditions were harvested from the matrices after 3 days before transferring to a low-attachment plate for suspension culture. **a** Live cell images of gel-embedded and suspended organoids at days 12 (D12) and 28 (D28); scale bar = 100 μm. Suspension of organoids was independently repeated twice. **b** Images of harvested organoids cut at 5 μm thickness and stained by haematoxylin and eosin; scale bar = 100 μm.

**c** Harvested organoids fixed, immunolabelled for syndecan-1 (SDC-1, pink) and co-stained with DAPI (grey). Confocal z-stacks of 60 μm depth were processed for visualisation using NIS Elements denoise.ai algorithm. Z-stack images of an organoid from each condition presented as maximum intensity projections (MIPs) with *XY* and *XZ* orthogonal views and optical z slices 30 μm apart. Arrows depict syncytialised areas on the outer surface of organoids. Scale bar = 100 μm. Suspension of organoids was independently repeated twice.

in the presence of TNF-α, although it reduced STBs in the absence of TNF-α. These differences with aspirin or metformin treatment in relation to the STB or EVT content of organoids could be dose- or context-dependant, as we have shown previously[36]. Additionally, pathway analysis of aspirin and metformin treatment in inflamed trophoblast organoids could provide further insights into the effect of

these agents on placental growth and development. Overall, we demonstrated that our automated, high-throughput and precise bioprinted trophoblast organoid model can provide a myriad of useful parameters to screen potential treatments for preeclampsia and accelerate drug discovery by investigating organoid number, growth, metabolism, proliferation, viability and differentiation. This is

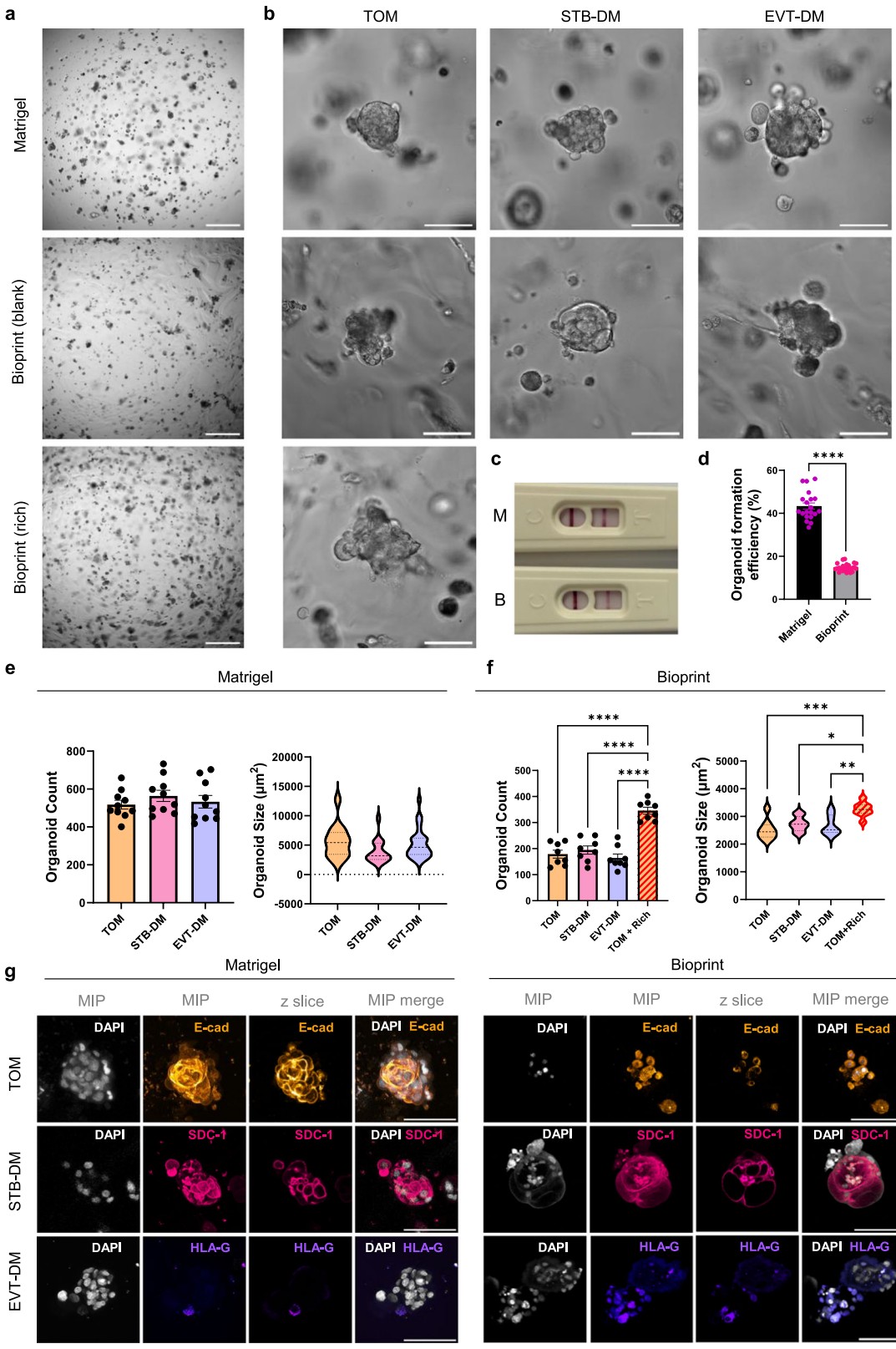

particularly important as preeclampsia is still a condition without a cure. The same platform can be used to extract mRNA and protein from bioprinted organoids and investigate mechanisms of placentation or mechanisms of action of potential treatments.

To mitigate the inside-out architecture of trophoblast organoids, we performed a suspension culture method like that applied in other organoid models[64–67]. Suspension of immature Matrigel or bioprinted

organoids resulted in STB localisation to the outer surface, detected by expression of SDC-1. Higher magnification imaging of suspended organoids also revealed microvilli-like structures on the periphery of SDC-1+ cells, potentially reflecting the brush border of the in vivo STBs. The reversal of trophoblast organoid polarity by suspension culture is in line with a recently published study, where primary trophoblast-derived organoids were grown in suspension with gentle agitation,

**Fig. 6 | Trophoblast stem cell organoid formation in Matrigel and bioprinted conditions.** Matrigel-embedded or bioprinted trophoblast stem cell (CT29) organoids supported by trophoblast organoid medium (TOM), syncytiotrophoblast differentiation medium (STB-DM) or extravillous trophoblast differentiation medium (EVT-DM). Bioprinted organoids were additionally generated in a rich PEG matrix. **a** Day 12 brightfield images, Nikon Ti, ×4, scale bar = 500 μm. **b** Day 12 brightfield images, Nikon Ti, ×20, scale bar = 100 μm. **c** Over-the-counter pregnancy test result of conditioned medium from STB-DM organoids in Matrigel (M) or bioprinted (B). **d** Organoid formation efficiency calculated as the number of cells seeded per unit/number of organoids. Unpaired, two-tailed *t*-test, data plotted as mean ± SEM, data points represent the mean of each bioprint unit, ****$p < 0.0001$.

**e** Number of organoids and organoid area of Matrigel-embedded organoids. Ordinary one-way ANOVA with Tukey's post-hoc, data plotted as mean ± SEM, data points represent the mean of each bioprint unit. Violin plots display median ± interquartile range (IQR) in organoid size across bioprint units. **f** Number of organoids and organoid area of bioprinted organoids. Ordinary one-way ANOVA with Tukey's post-hoc, data points represent the mean of each bioprint unit. Violin plots display media ± IQR in organoid size across bioprint units. *$p < 0.05$, **$p < 0.01$, ***$p < 0.001$, ****$p < 0.0001$ (Blank vs. Rich $p = 0.0005$; STB-DM vs. Rich $p = 0.0144$; EVT-DM vs. Rich $p = 0.0072$). **g** Immunofluorescence images of organoids labelled for key trophoblast subtype markers. Leica Stellaris confocal microscope, 25x, scale bar = 100 μm. Source data are provided as a Source Data file.

resulting in consistent layers of STBs on their outer surface[68]. Orbital rotation was also proposed to improve polarity reversal by inducing gentle shear stress that mimics early placental physiology and promotes trophoblast fusion[69]. Fluid flow in trophoblast organoid culture could also improve the viability of suspended organoids for long-term culture by facilitating nutrient permeation through to the core and preventing necrosis[70].

Finally, we tested TSC organoid formation in Matrigel compared to a bioprinted PEG matrix. It was interesting, though somewhat unsurprising, to observe reduced organoid formation and growth of TSCs in the stiffer matrix. Given that the endometrium is reduced in stiffness during the secretory phase of the menstrual cycle and decidualisation during the window of implantation, we expect that TSCs arising from this early blastocyst stage would require a softer environment for expansion[71,72]. Additionally, the secretory phase is marked by altered expression of ECM proteins, with the upregulation of collagen IV and V, laminin and fibrillin[73–75], which is aligned with TSC organoids showing improved growth profile in the 'rich' matrix in our bioprinted model. This highlights an important distinction in the ECM environment requirements of TSCs in the stem cell niche and their mature progeny, such as ACH-3Ps. Since increased placental stiffness has been observed in tissue from pregnancy complications, a system that supports screening of a multitude of matrix stiffnesses and proteins would enable further investigation into these crucial mechanisms.

To extend this model, the matrix stiffness could be tuned to induce changes consistent with gestational age or pregnancy disorders, including preeclampsia, foetal growth restriction and gestational diabetes mellitus[76–82]. Organoid proliferation and cell fate may be further dictated by introducing adhesion factors required for growth and differentiation. To advance this bioprinted model of placental organoids to investigations of maternal-foetal interactions, uterine endothelial cells, stromal cells, fibroblasts or immune cells (e.g. decidual natural killer cells or macrophages) can be incorporated. Additionally, while most cell cultures are maintained at 21% oxygen, the early placenta develops within a physiologically low oxygen environment of around 2.5%[83]. Given that stem cell niches also often reside in 2–8% oxygen, culture in low oxygen conditions may improve the expansion of organoids and their biological relevance[84]. While we acknowledge that a limitation of this study is the influence of choriocarcinoma within the ACH-3P cell line, we have demonstrated at the transcriptomic level that ACH-3P bioprinted organoids closely resemble human first-trimester placental tissue. Also, we have shown the application of our approach to biofabricating TSC organoids that need further matrix tuning, perhaps using a softer matrix, in the future. This study has demonstrated that ACH-3P organoids can be used as a low-cost, low-risk, reproducible and biologically relevant model to supplement the use of primary cells that require intensive sampling and culture conditions with expensive components. Likewise, PEG matrix formulations open opportunities to control organoid formation and differentiation, with a 1.1 kPa blank matrix suitable for studies of placental EVT formation and invasion. Further, the application of

bioprinting facilitates accurate orientation of low volumes of PEG components, ensuring precise cross-linking and reproducible organoid formation. This bioprinted organoid model can therefore be used to study important physiological processes, pathogenic mechanisms driving impaired EVT differentiation and invasion, and assess potential therapeutics at low cost and risk. This opens new avenues to assess the molecular pathways involved in EVT differentiation by removing the need for exogenous chemical stimuli to drive differentiation. At the same time, we have shown that suspended organoid cultures offer the opportunity to more accurately study STB physiology, including barrier effects and syncytial shedding of nuclear aggregates or extracellular vesicles for the diagnosis of pregnancy complications[60,61,85–88].

## Methods

The use of ACH-3Ps and TSCs in this study was approved by the University of Technology Sydney Human Research Ethics Committee in accordance with the requirements of the NHMRC National Statement on Ethical Conduct in Human Research (2007).

### Cell culture

The ACH-3P first-trimester trophoblast cell line was generously donated by Professor Gernot Desoye (Graz Medical University, Austria), which was established in 2007[22]. ACH-3Ps were immortalised by the fusion of primary first-trimester trophoblasts isolated from a 12-week gestation placenta and choriocarcinoma cell line AC1-1, resulting in a hybrid male cell line. This cell line was chosen due to its cytotrophoblast-like phenotype and demonstrated capacity to form HLA-G⁻ and HLA-G⁺ subtypes, suggesting differentiation potential[22,89]. ACH-3P cells were cultured in Ham's F12 nutrient mix (Gibco; Thermo Fisher Scientific, cat. 11765062) supplemented with 10% foetal bovine serum (Gibco; Thermo Fisher Scientific, cat. 10099141) and 1% penicillin–streptomycin (Gibco; Thermo Fisher Scientific, cat. 15140-122). Every 2–5 passages, cells were treated with a selection medium containing azaserine (5.7 μM, Sigma-Aldrich) and hypoxanthine (100 μM, Sigma-Aldrich) for 24 h to prevent the overgrowth of choriocarcinoma. Cells were dissociated using Accutase (Sigma-Aldrich, cat. A6964) and experiments were performed with passages 13–15.

A male TSC ("Okae") line, CT29, was obtained from Riken (Japan). Culture surfaces were pre-coated with 0.5 μg/ml iMatrix-511 (FUJIFILM Wako, cat. 385-07361). CT29 cells were maintained in TSC 2D medium consisting of DMEM/F12 with 2.5 mM L-glutamine (Thermo Fisher Scientific, cat. 11320033), 0.15% bovine serum albumin (BSA, Sigma-Aldrich, cat. A8806), 1% insulin–transferrin–selenium–ethanolamine (ITS -X, Sigma-Aldrich, cat. I3146), 1% KnockOut™ serum replacement (KSR, Thermo Fisher Scientific, cat. 10828010), 0.2 mM L-ascorbic acid (FUJIFILM Wako, cat. 1713265-25-8), 0.5% penicillin–streptomycin (P/S, Thermo Fisher Scientific), 2.5 μM Y27632 (Selleck Chemicals, cat. S6390), 25 ng/ml epidermal growth factor (EGF, FUJIFILM Irvine Scientific, cat. 100-26), 0.8 mM valproic acid (VPA, Selleck Chemicals, cat. 99-66-1), 1 μM A83-01 (Selleck Chemicals, cat. S7692), 2 μM CHIR99021 (Selleck Chemicals, cat. S1263) and SB431542 (Selleck Chemicals, cat. S1067)[53]. Organoids were generated using passage 24.

Cell cultures were incubated at 37 °C, 5% CO$_2$ in a humidified atmosphere and were routinely tested for mycoplasma contamination.

## Generation of Matrigel-embedded organoids

To prevent two-dimensional cell attachment, the wells of a black-walled, 96-well plate (Perkin Elmer, cat. 6055300) were precoated with 15 μL of 0.5% agarose (Sigma-Aldrich, cat. A9539). Dissociated cells were filtered through a 70 μm filter to separate aggregates before 500 ACH-3P cells were seeded in 20 μL droplets or 1200 TSC cells were deposited in 5 μL droplets using cold, growth-factor reduced Matrigel (Corning, cat. CLS356231). The cell-laden Matrigel droplets were incubated at 37 °C for 40 min to polymerise prior to the addition of 150 μL ACH-3P growth medium or TSC organoid medium (TOM), respectively. TOM contained DMEM/F12 with 2.5 mM ʟ-glutamine (Thermo Fisher Scientific, #11320033), 0.15% bovine serum albumin (BSA, Sigma-Aldrich, #A8806), 1% insulin−transferrin−selenium−ethanolamine (ITS-X, Sigma-Aldrich, #I3146), 1% KnockOut™ serum replacement (KSR, Thermo Fisher Scientific, #10828010), 0.2 mM ʟ-ascorbic acid (FUJIFILM Wako, #1713265-25-8), 0.5% penicillin−streptomycin (P/S, Thermo Fisher Scientific), 2.5 μM Y27632 (Selleck Chemicals, #S6390), 25 ng/mL epidermal growth factor (EGF, FUJIFILM Irvine Scientific, #100-26), 1 μM A83-01 (Selleck Chemicals, #S7692) and 2 μM CHIR99021 (Selleck Chemicals, #S1263). Matrigel cultures were grown for 12 days, with culture medium changed every 3–4 days in ACH-3P cultures or 2–3 days in TSC cultures.

## Generation of bioprinted organoids

Large Plug 3D cell models (7.2 μL/well) were generated using the RASTRUM drop-on-demand bioprinter (Inventia Life Science). The wells of a black-walled, 96-well plate were initially printed with an Inert Base to prevent 2D attachment. Dissociated and filtered ACH-3P cells were primed and printed at 10,000 cells/well in a synthetic PEG-based matrix (Inventia Life Science, cat. Px02.00). The system uses a two-droplet process, with the PEG-4MAL bioink meeting the cells suspended in an MMP-cleavable activator that immediately polymerises as the two active bioink components make contact and crosslink[90]. The RASTRUM matrix formulations used were all 1.1 kPa stiffness, resembling that of the decidua basalis, the site of placental invasion. A matrix selection was performed to compare (i) a naked 'blank' PEG containing no additional adhesion peptides, (ii) a matrix containing α- and β-laminin peptide chains (IKVAV and YIGSR), and (iii) a matrix containing collagen type I peptide mimetic (GFOGER) given the high expression of laminin and collagens in the native placenta[75,91–93]. Bioprinted TSC organoids were produced in Large Plug models in either naked PEG, as used for ACH-3P, or a 'rich' matrix of ~1.1 kPa and containing RGD (fibronectin, laminin), YIGSR (laminin) and CNYYSNS (collagen IV) adhesion peptides with laminin-511 and hyaluronic acid (Inventia Life Science, cat. Px02.72). These cells were printed at a seeding concentration of 2,780,000 cells/mL which equated to a density of 10,000 cells/well. Bioprinted cultures were immediately overlaid with ACH-3P or TSC organoid growth medium and grown for 12 days, with culture medium changed every 3–4 days for ACH-3P or 2–3 days for TSCs.

## Differentiation of TSC organoids

TSC organoids were differentiated from day 7 using either STB or EVT differentiation medium (STB-DM and EVT-DM, respectively). STB-DM contained DMEM/F12 with 2.5 mM ʟ-glutamine (Thermo Fisher Scientific, cat. 11320033), 0.3% bovine serum albumin (BSA, Sigma-Aldrich, cat. A8806), 1% insulin−transferrin−selenium−ethanolamine (ITS-X, Sigma-Aldrich, cat. I3146), 4% KnockOut™ serum replacement (KSR, Thermo Fisher Scientific, cat 10828010), 0.2 mM ʟ-ascorbic acid (FUJIFILM Wako, cat. 1713265-25-8), 0.5% penicillin−streptomycin (P/S, Thermo Fisher Scientific), 2.5 μM Y27632 (Selleck Chemicals, cat. S6390), 50 ng/mL epidermal growth factor (EGF, FUJIFILM Irvine Scientific, cat. 100-26) and 2 μM forskolin (FUJIFILM Wako, cat. 067-02191). EVT-DM contained DMEM/F12 with 2.5 mM ʟ-glutamine (Thermo Fisher Scientific, cat. 11320033), 0.3% bovine serum albumin (BSA, Sigma-Aldrich, cat. A8806), 1% insulin−transferrin−selenium−ethanolamine (ITS-X, Sigma-Aldrich, cat. I3146), 4% KnockOut™ serum replacement (KSR, Thermo Fisher Scientific, cat. 10828010), 0.5% penicillin−streptomycin (P/S, Thermo Fisher Scientific), 2.5 μM Y27632 (Selleck Chemicals, cat. S6390), 7.5 μM A83-01 (Selleck Chemicals, cat. S7692) and 100 ng/mL neuregulin-1 (Antibodies.com, cat. A63582-10).

## Bioprinted organoid drug screening

ACH-3P cells were bioprinted into a blank PEG matrix at a cell density of 8,000-14,000 cells/well. Following bioprinting, 150 μL of ACH-3P growth medium was added to each well, and the plate was incubated at 37 °C with 5% CO$_2$. To assess the impact of TNFα concentration on bioprinted organoids, we first exposed them to varying concentrations (0, 1, 5 and 20 ng/mL) from day 5 of culture. For subsequent experiments, TNFα (20 ng/mL) was added to respective wells from day 5 of culture. Aspirin (0.5 mM) and metformin (0.5 mM) were added to respective wells from day 8 of culture. Imaging and downstream analysis were conducted as described below.

## Organoid suspension to invert polarity

After three days in matrix-embedded culture, wells were washed with phosphate-buffered saline (PBS) and organoids from both Matrigel and bioprinted conditions were retrieved from their matrices. To retrieve organoids from Matrigel, 100 μL of 1 U/mL dispase (StemCell Technologies, cat. 07913) was added to each well and incubated for 30 min at 37 °C until gel dissolved. Bioprinted organoids were retrieved from their matrix by incubating each well with 75 μL of Cell Retrieval Solution (Inventia Life Science) for 30 min at 37 °C. Matrix digestion was halted by adding complete growth medium, well contents were collected and washed twice with PBS. Organoids were resuspended in complete growth medium and transferred to an ultra-low attachment 24-well plate (Corning, United States). Organoids were maintained in suspended culture for up to either 12 or 28 days with medium changed every 3–4 days.

## Live cell imaging

Live cell imaging was performed using an IncuCyte organoid assay module (Sartorius, Germany) with ×10 brightfield and phase images acquired every 6 h per well. Higher resolution brightfield images were acquired with a Nikon Ti inverted microscope (Nikon, Japan) using a Nikon DS-Qi2 camera, NIS Elements Advanced Research software (v. 5.30.06), ×4 (Plan Apo λ, NA 0.2), ×10 (Plan Fluor Ph1 DLL, NA 0.3), and ×20 (Plan Apo λ ×20 Ph2 DM, NA 0.75) objectives.

Organoid count was taken at day 7 in Matrigel versus bioprint experiments to account for progressive fusion of organoids. Organoid area was measured using ×10 magnification images acquired from 5 wells of each experimental replicate, totalling at least 800 organoids per matrix condition.

## Viability staining

Cell viability was assessed in Matrigel and bioprinted models using a live/dead cell viability kit for mammalian cells (Biotium, cat. 30002). Encapsulated cells were washed with PBS and incubated with calcein-AM (1 μM), ethidium homodimer 3 (EthD-III, 2 μM), and Hoechst 33342 (10 μg/mL; Invitrogen, cat. H3570) in PBS to label live cells, dead cells and nuclei, respectively. Wells were incubated with the fluorescent dyes for 30 min at 37 °C and washed twice with PBS prior to imaging. Images were acquired at ×10 magnification using the IncuCyte green and red channels for live and dead cells, respectively. Higher resolution representative fluorescent images were acquired as a z-stack of 20 slices at 1 μm step size using a Leica Stellaris confocal microscope.

Images were acquired using a ×20 objective (HC PL APO CS2 ×20, NA 0.75), DAPI (ex 450 nm), FITC (ex 495 nm) and TRITC (ex 590 nm) channels and a pinhole size of 66.7 μm at 590 nm. Wells containing Matrigel or inert base only, calcein-AM only or EthD-III only were used as staining controls.

## Alamar Blue proliferation assay

Alamar Blue (Invitrogen, cat. DAL1100) was prepared in complete cell culture medium (10%) and added to culture wells on days 1, 4, 7 and 11 in Matrigel versus bioprinted conditions or days 6, 9 and 12 in the drug screening assay. After 2 h incubation, the metabolised Alamar Blue medium was transferred to a new plate and the fluorescent intensity at 530 nm excitation and 590 nm emission was recorded using a Tecan M Plex plate reader. At least 3 wells were measured per time point per condition.

## C-reactive protein ELISA

Organoid growth medium from the TNF-α concentration curve and drug screen assays was collected at day 12. Medium from the 3 wells of each condition was pooled and assessed using a CRP ELISA (Abcam, cat. ab260058) as a measure of inflammation[94]. Experiments were conducted according to the manufacturer's instructions.

## Organoid harvesting

Matrigel-embedded organoids were harvested using dispase enzyme, and bioprinted organoids were collected using Cell Retrieval Solution as described above. Retrieved organoids from both conditions were washed twice with PBS to remove residual matrix prior to downstream analysis.

## Immunofluorescent labelling

A whole organoid immunolabelling protocol was adapted from Rios et al.[95]. Harvested or in situ ACH-3P organoids and in situ TSC organoids were fixed with 4% paraformaldehyde (PFA; Sigma-Aldrich, P6148) for 1 h at 4 °C and washed three times with PBS. When immunolabelling for intracellular proteins (β-hCG and Cyk-7), organoids were first permeabilised for 30 min at room temperature in a penetration buffer (PBS, 20% DMSO, 0.3 M glycine, 0.2% Triton X-100). Permeabilised organoids were washed twice with organoid washing buffer (OWB; containing PBS, 1% bovine serum albumin (BSA), 0.1% Triton X-100) for 5 min at room temperature. All organoids were incubated in blocking buffer (PBS, 1% bovine serum albumin, 3% normal goat serum, 0.1% Triton X-100) for 1 h at 4 °C by gentle rocking. Organoids were incubated with primary antibodies prepared in blocking buffer according to dilutions (Supplementary Table 1) overnight at 4 °C with gentle rocking. Primary antibodies were removed, and organoids were washed three times with OWB for 2 h at 4 °C with gentle rocking. Secondary antibodies were prepared in blocking buffer and incubated with organoids overnight at 4 °C with gentle rocking. Secondary antibodies were removed, and organoids were washed three times with OWB for 2 h at 4 °C with gentle rocking. OWB was removed and organoids were optically cleared using a fructose (2.5 M)–glycerol (60% v/v) clearing solution, incubated at room temperature for 20 min and transferred to a microscope slide. Organoids incubated with secondary antibodies only were used as labelling controls.

Fluorescent images of mounted ACH-3P organoids were acquired using a Nikon A1R inverted confocal microscope with PMT and GaAsP detectors and NIS Elements Advanced Research software (v. 5.30.06). Z-stacks taken at 1 μm slices to a depth of 60 μm were acquired by Nyquist sampling for each organoid using a ×20 long working distance objective (S Plan Fluor LWD, NA 0.7). Laser excitation lines of 405, 488, 561 and 637 nm were used with a line average count of 2 and 33.2 μm pinhole diameter. Higher resolution images were also acquired with a ×100 oil immersion objective (Plan Apo λ, NA 1.45). Representative

images were processed using NIS Elements denoise.ai. Fluorescent images of in situ ACH-3P or TSC organoids were acquired using a Leica Stellaris (LASX 4.8.1) inverted confocal microscope with HyD X and HyD S detectors, white light laser and a ×25 water immersion objective (NA 0.95). Z stacks were taken at 1.5 μm slices with a line accumulation count of 2 and a pinhole of 55.82 μm (1 Airy unit).

## Image analysis

Organoid viability was calculated using IncuCyte software (v2022B.rev2) by applying a mask to the areas occupied by green or red fluorescence. Viability data were presented as an area positive for calcein-AM (live)/total cell area (live + dead). Maximum intensity projections were created using ImageJ (v1.53j; https://imagej.net/software/fiji/). The proportion of cells positive for HLA-G and β-hCG was assessed using Bitplane Imaris (v9.6.0) software. At least 10 confocal z stacks were processed using Bitplane Imaris Surfaces module to quantify nuclei and respective immunofluorescent markers for each condition and experimental repeat (n = 10–92 per experimental replicate).

## Organoid processing for paraffin embedding

Organoids and bioprinted matrix plugs harvested for histology were fixed in 4% PFA at room temperature for 1 h. Organoids were washed with PBS and stored in 70% ethanol overnight at 4 °C. The ethanol was removed, organoids were suspended in approximately 100 μL molten 1% low-gelling temperature agarose (Sigma-Aldrich, A4018) and transferred to a plastic surface to set as a droplet. After 20 min, the agarose droplet was transferred to a histology cassette and processed through the following for 45 min each: 80% ethanol, 90% ethanol, 100% ethanol, 100% ethanol, xylene, xylene, molten paraffin wax. The wax-perfused agarose droplets were then embedded in paraffin wax for microtome sectioning at 5 μm thickness.

## Haematoxylin and eosin staining

Dried microscope slides containing sections of organoid droplets were passed through the following reagents for 2 min each, unless otherwise stated: xylene (3 times), 100% ethanol (3 times), 95% ethanol, 70% ethanol, water, Harris Haematoxylin (5 min), water, acid alcohol (15 s), water, Scott's Bluing solution, water, 1% alcoholic Eosin (3 min), 100% ethanol (3 times), xylene (3 times). Coverslips were applied using DPX mounting medium (LabChem, AJA3197) and images acquired at ×20 (NA 0.50) using an Olympus BX51 upright microscope.

## RNA extraction and RT-qPCR

To isolate RNA, harvested organoids were washed twice with PBS before extraction with 1 mL QIAzol (QIAGEN, 79306) according to the manufacturer's protocols. RNA lysates were assessed using a Nano-Drop One spectrophotometer (Thermo Fisher Scientific) and gene expression quantified by RT-qPCR using a Luna Universal One-Step kit (New England Biolabs, E3005) and C1000 Touch Thermal Cycler (Thermo Fisher Scientific). Relative gene expression was quantified using the $2^{-\Delta\Delta Ct}$ method. The primers were purchased from Sigma-Aldrich: β-actin (#NM_001101; forward: GACGACATGGAGAAAATCTG, reverse: ATGATCTGGGTCATCTTCTC); Il-6 (#NM_000600; forward: GCAGAAAAAGGCAAAGAATC, reverse: CTACATTTGCCGAAGAGC).

## Single-cell RNA sequencing

For single-cell RNA sequencing (scRNA-seq), 7 wells of each condition were harvested and pooled, producing roughly 700 organoids. Organoids were washed twice with PBS and centrifuged for 5 min at 300 × g. Organoids were dissociated into single cells by adding 1 mL Accutase to each sample, followed by incubation on a shaker for 30 min at 800 rpm. The cell suspension was centrifuged for 5 min at 300 × g and washed twice with sterile 0.04% BSA in PBS, centrifuging between each step. Cells were then resuspended in 100 μL 0.04% BSA and passed

through a 40 μm filter. Cell viability and count were assessed using trypan blue, ensuring viability was >80%.

Single cells were prepared for sequencing according to the 10× Genomics Chromium Single Cell 3' Kit user guide at the Single Cell Technology Facility at the Faculty of Engineering and IT at UTS. Briefly, to create Gel Beads-in-emulsion (GEMs), approximately 5000 single cells from each sample were suspended in a Chromium master mix and combined with barcoded Single Cell 3' v3.1 Gel Beads and partitioning oil was loaded onto a Chromium Next GEM Chip G for partitioning. In each resulting GEM, the Gel Bead was dissolved and co-partitioned with cell lysed, incubated with reverse transcription reagents and a unique molecular identifier (UMI) to produce barcoded, full-length cDNA. Barcoded, full-length cDNA was amplified via 12 PCR cycles to generate sufficient mass for library construction. A sample index PCR was performed to generate a 3' gene expression dual index library, which was double-sided size selected. Quality control was performed using an Agilent TapeStation and analysed using TapeStation Analysis Software (v4.1.1). The libraries were sequenced by the Australian Genome Research Facility (AGRF) on an Illumina NovaSeq 6000 instrument using a NovaSeq S4 kit (200 cycles) v1 chemistry at an aimed depth of 30,000 paired-end reads per cell. The sequencer generated raw data files in binary base call (BCL) format.

### Single-cell RNA-seq data pre-processing, quality control and analysis

The BCL files were demultiplexed and converted to the FASTQ file format using Illumina Conversion Software (bcl2fastq v2.19.0.316). The 10x Genomics cellranger-v (3.1.0) count pipeline was used for alignment, filtering, barcode counting, and UMI counting from FASTQ files and was executed on a high-performance cluster with 2.6.32-754.17.1.el6.x86_64 operating system.

The single-cell datasets were analysed with Seurat. Cells with at least 100 genes and <15% mitochondrial genes were kept. Genes expressed in fewer than 5 cells were discarded. The 2000 most variable genes were used in PCA calculations. Cells were analysed with 30 principal components (PCs) and clustered with 0.2 and 0.3 resolution. DEG between clusters for each compartment were calculated with FindMarkers with min.pct = 0.25 and logfc.threshold = 0.4 using the Wilcoxon algorithm. Gene set enrichment analysis of cluster-specific markers was performed with the fgsea package. Reactome or Gene Ontology gene sets were downloaded from the MSigDB and obtained from "h.all.v2023.2.Hs.entrez". The Monocle3 R package was used to explore the differentiation of ACH-3P trophoblasts. The raw counts were loaded into Monocle3 for semi-supervised single-cell ordering in pseudo-time, using SPINT1 expression to denote the origin. Branched expression analysis modelling (BEAM) was applied to observe the expression of the CTB progenitor gene markers along pseudo-time trajectory towards either STB or EVT.

To quantify how closely our organoid trophoblast lineages resemble their in vivo counterparts, we benchmarked scRNA-seq profiles from Matrigel- and bioprint-derived organoids against three reference datasets: (i) a first-trimester (8–12 weeks) chorionic-villus atlas[28], (ii) a 6-week placental trophoblast dataset[29], and (iii) a published patient-derived trophoblast organoid dataset[34]. All raw count matrices were processed in Seurat v4.4 with log-normalisation. Cells annotated as CTB, EVT or STB were subset. Organoid cells were labelled by culture platform (Matrigel, Bioprint), and public cells retained their native study identifiers at the resolution of only CTB, EVT and STB. SCE objects were merged, and MetaNeighbor[35] was applied to identify the set of study-invariant, highly variable genes (variableGenes function) and to compute pairwise area-under-the-ROC (AUROC) similarity scores between every cell-type–by-study pair using the fast_version = TRUE implementation. AUROC values range from 0.5 (random overlap) to 1 (perfect concordance); we considered scores > 0.88 as evidence of strong transcriptional similarity. Results were

visualised as a similarity network (igraph v1.6.0, ggraph v2.2.1) in which edges connecting two nodes (cell-type-by-study labels) were drawn only when AUROC > 0.88, and node size was scaled to the within-dataset self-AUROC. The analysis was repeated with organoid cells pooled (Matrigel + Bioprint) and with each platform analysed separately to assess platform-specific fidelity (Supplementary Fig. 2).

### Proteomics sample preparation

For proteomic analysis, 8 wells of each condition were harvested as described above and pooled, amounting to approximately 800 organoids per sample for each of three experimental repeats. Harvested organoids were washed twice with PBS, PBS was removed, and the dry organoid pellet was stored at −80 °C prior to downstream analysis. Undiluted Matrigel (20 μL) and ACH-3P cells grown in 2D were prepared as controls. Organoid pellets were thawed and solubilised with 30 μL SDC Master Mix (Supplementary Table 2) at 95 °C for 10 min. Samples were cooled at room temperature for 50 min prior to the addition of 2 μL trypsin solution and overnight incubation at 37 °C. Load Buffer was added to each sample (300 μL) and samples were centrifuged for 5 min at 15,000 × g. Stage tips were prepared using styrene divinylbenzene- reverse phase sulfonate (SDB-RPS), and 165 μL of supernatant was transferred to each tip[96]. Tubes with stage tips were centrifuged for 2 min at 2100 x g. Stage tips were washed with 100 μL Wash Buffer and centrifuged for 2 min at 2100 × g. Stage tips with bound peptides were transferred to new tubes containing inserts (Agilent), 50 μL fresh Elution Buffer was added to each sample and then centrifuged for 2 min at 2100 × g. Insets containing eluted peptides were evaporated in a centrifugal vacuum concentrator for 1 h.

### Liquid chromatography/mass spectrometry/mass spectrometry (LC/MS/MS)

Peptides were resuspended in 25 μL MS Loading Solvent. Using an Acquity M-class nanoLC system (Waters, USA), 5 μL of the sample was loaded at 15 μL/min for 3 min onto a nanoEase Symmetry C18 trapping column (180 μm × 20 mm) before being washed onto a PicoFrit column (75 μm ID × 350 mm; New Objective, Woburn, MA) packed with SP-120-1.7-ODS-BIO resin (1.7 μm, Osaka Soda Co, Japan) heated to 45 °C. Peptides were eluted from the column and into the source of a Q Exactive Plus mass spectrometer (Thermo Scientific) using the following programme: 5−30% MS buffer B (98% Acetonitrile + 0.2% formic acid) over 90 min, 30−80% MS buffer B over 3 min, 80% MS buffer B for 2 min, 80−5% for 3 min. The eluting peptides were ionised at 2400 V. A data-dependant MS/MS (dd-MS2) experiment was performed, with a survey scan of 350–1500 Da performed at 70,000 resolution for peptides of charge state 2+ or higher with an AGC target of 3e6 and maximum Injection Time of 50 ms. The Top 12 peptides were selected and fragmented in the HCD cell using an isolation window of 1.4$m/z$, an AGC target of 1e$^5$ and a maximum injection time of 100 ms. Fragments were scanned in the Orbitrap analyser at 17,500 resolution and the product ion fragment masses measured over a mass range of 50–2000 Da. The mass of the precursor peptide was then excluded for 30 s. LC/MS/MS was performed on each sample in triplicate injections.

### Mass spectrometry data analysis

The MS/MS data files were searched using Peaks Studio 11 against the UniProt Human reference proteome database (20230323) and a database of common contaminants with the following parameter settings. Fixed modifications: none. Variable modifications: propionamide, oxidised methionine, deamidated asparagine. Enzyme: semi-trypsin. Number of allowed missed cleavages: 3. Peptide mass tolerance: 10 ppm. MS/MS mass tolerance: 0.05 Da. The results of the search were then filtered to include peptides with a −log10$P$ score that was determined by the false discovery rate (FDR) of <1%, the score being that where decoy database search matches were <1% of the total

matches. The protein.db file was exported from peaks, followed by a low-value imputation of 5000, representing a low abundance area value for any value that had a numerical value of 0. This approach was used in favour of other methods that use random values within the distribution, or values reduced by factors of the standard deviation. This dataset had a wide distribution of area values across several orders of magnitude due to large changes in protein abundance between the different experimental groups. This meant that reduction by SD or assigning random values within the distribution was inappropriate and sometimes resulted in negative values. For this reason, a representative value of 5000 was chosen as this was similar to other very low abundance proteins. Student's $t$-tests were used between comparable groups. A log2-fold change of $<-0.4$ or $>0.4$ with a $p$-value of $<0.05$ was considered significant and included in downstream analyses. Significantly differentially abundant proteins were searched in Reactome.org. Due to the limitations of the Reactome database and our sample type not being very well represented in the pathway analysis, a $p$-value of $<0.05$ was used as a cutoff for significantly enriched pathways instead of a more stringent FDR of 1%. Significantly enriched pathways were then further interrogated in the literature to validate that the record in the database was correct.

## Statistical analysis

Organoid data were statistically analysed using GraphPad Prism (v9.4.1). For normally distributed data, an unpaired, two-tailed $t$-test was used to compare the means of experimental replicates, where 2 groups were considered, and a $p$-value $< 0.05$ was considered statistically significant. Where more than one condition was assessed, an ordinary one-way ANOVA was applied with either Tukey's or Šídák's multiple comparisons post-hoc test. Data are presented as mean ± standard error of the mean (SEM).

## Reporting summary

Further information on research design is available in the Nature Portfolio Reporting Summary linked to this article.

## Data availability

The raw single-cell RNA-sequencing data for the trophoblast organoids have been deposited in the GEO under accession number GSE279994. All protein mass spectrometry raw data is available in a ProteomeXchange partner repository with identifier PXD056796. Large raw image/micrograph files that support the findings in this study are not provided with the source data and are available from the corresponding author upon request. To request the data, please contact A/Prof Lana McClements at lana.mcclements@uts.edu.au. Requests will be evaluated and responded to within 10 business days. The data will be shared from the institutional repository and available for five years from the date of publication. Source data are provided with this paper.

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

## Acknowledgements

The authors would like to acknowledge the Single Cell Technology Facility, Faculty of Engineering and Information Technology, University of Technology Sydney. We would like to thank Holly Pearson for their scientific input and technical assistance. The authors acknowledge the use of the Nikon Ti widefield, Nikon A1R confocal, Leica Stellaris confocal and Olympus BX51 microscopes and Bitplane Imaris software at the Microbial Imaging Facility at AIMI in the Faculty of Science, the University of Technology Sydney. The authors acknowledge the Proteomics, Lipidomics and Metabolomics Core Facility for access to proteomics methodologies and instrumentation, as well as technical assistance. This research was supported by UTS Research Excellence Scholarship (C.R.), Australian Government Research Training Programme Scholarship (C.R. and A.B.) and Seed Funding provided by the Faculty of Science, UTS (C.R.). H.C. was supported by the UTS President's Scholarship and International Research Training Programme (IRTP) Scholarship. L.M. was supported by a Future Leader Fellowship Level 1 (106628) from the National Heart Foundation of Australia.

## Author contributions

C.R. conceptualised, optimised, and carried out the experiments, analysed and interpreted the data and wrote the manuscript. H.C. analysed data, contributed to data interpretation and manuscript writing. M.O. and M.P.P. contributed to proteomics experiments, data analysis and interpretation. Ash.B. and G.O. performed experiments and data analysis. A.V., A.G., and K.M. contributed to experimental design and data interpretation. C.G. provided histology training, assisted in experimental design and optimisation. D.G.O. contributed to experimental design, technical support and data acquisition through the Single Cell Technology Facility. A.B and L.C. provided significant input to experimental design and optimisation, image acquisition, image analysis and interpretation. P.H. supervised H.C. and contributed to data analysis and interpretation. L.M. conceptualised the study, acquired funding, contributed to data analysis and interpretation and manuscript writing. L.M. and K.M. supervised C.R. All authors approved the final version of the manuscript.

## Competing interests

A.V. was an employee of Inventia Life Science at the time of the study. The remaining authors declare no competing interests.
