## [Transparent Peer Review file · Nature Communications]

Matrix directs trophoblast differentiation in a bioprinted organoid model of early placental development

Corresponding Author: Professor Lana McClements

Version 0:

Reviewer comments:

Reviewer #1

(Remarks to the Author)

The manuscript by Richards et al set out to test organoid forming ability of the trophoblast/choriocarcinoma hybrid cell line in a synthetic bioprinted PEG scaffold. The rationale of the study centers on the valid concern on the reliance of the extracellular matrix substance, Matrigel. Derived from rodent sarcoma, Matrigel has significant batch variability and the composition of the proteins are not entirely well defined. This inconsistency has undoubtedly contributed to discrepancies in scientific findings. Therefore the underlying goal of the work to examine the feasibility of implementing a synthetic PEG-based scaffold for organoid culture is valid and important.

I had difficult time reviewing this manuscript as there seemed to be missing components in the results section. But as written, I had a difficult time following the logic of the specific extracellular matrix proteins selected for PEG bioprinting. Moreover, I was unsure which protein was selected for PEG bioprinting; based on the discussion it appears the authors performed most if not all their characterizations using the naked PEG scaffold. However, as indicated, this was not clear within the manuscript.

In addition to not understanding the flow/logic of the paper, I also have concerns related to the use of the ACH-3P cell line as a trophoblast organoid model. While not entirely novel (as the authors rightly indicate; Dietrich et al, Placenta 2023 PMID: 36696785), the authors chose to use a cell culture media that is somewhat ill-defined and that does not promote conditions of trophoblast regeneration or stemness. This decision limits how the current model can be directly compared to current state-of-art trophoblast organoid systems, and importantly, dampens the potential impact of the manuscript as an application to progenitor/stem cell-based trophoblast organoid systems.

Below are my specific comments:

1. Figure 1 is present in the manuscript body, but prose in the results related to Figure 1 are missing.
2. Relatedly, there appears to be missing text related to the logic and choice of specific ECM proteins being incorporated into PEG hydrogels. As such, it was not possible to adjudicate the upfront justification of the use of naked PEG for all downstream analyses. This is a major limitation of the current manuscript as this is the major novel component of the work and likely the aspect of the work that could be highly valuable to the field.
3. This reviewer really questions the use of the culture conditions chosen (Hams F12 with 10% FBS) along with the ACH-3P cell line. While the justification of using an immortalized cell line could be made, the authors didn't really make it clear why this particular cell line was chosen for organoid forming abilities. The cell line used spontaneously differentiates into SCT- and EVT-like cells. This is an issue for the utilization of this model for studying trophoblast differentiation as the system lacks control and the differentiation kinetics are somewhat random.
4. Another limitation of the manuscript was the complete lack of comparison of ACH-3P data to in vivo. Single cell datasets of first trimester chorionic villi exist and would greatly improve the characterization of the ACH-3P model and importantly, the impact of different matrices on trophoblast biology. As such, it is difficult to adjudicate how the synthetic hydrogel impacts trophoblast regeneration and differentiation without comparison to a gold standard control.

5. Related to comment 3 above, I had a difficult time understanding why the authors did not examine how their PEG bioprint conditions impacted TSC organoid forming abilities, as TSC lines (Okae lines) are the current gold standard in trophoblast biology research due to their commercial availability and ease of use. While there are geographic locations that prohibit the use of TSCs, iPSC-derived TSCs (PMID: 35523141) and iTSCs (PMID: 32939092) bypass these restrictions. I feel the authors needed to make a strong case for focussing on the ACH-3P line, as it is arguably a lesser cell model for studying trophoblast biology.

Reviewer #2

(Remarks to the Author)

In this study, the authors 3D bioprinted trophoblast organoids and compared them with conventional Matrigel embedding trophoblast organoids. They used the trophoblast cell line ACH-3P encapsulated in poly(ethylene glycol) (PEG) bioink. They observed that ACH-3P in Matrigel or in PEG spontaneously formed organoids after 4 days of culture and these organoids were metabolically active and cells proliferation increased with time. The organoids were formed by cytotrophoblasts (CTs) with a central cavity, while the surface of the organoids was covered with extravillous trophoblasts (EVTs) with some clusters of syncytiotrophoblasts (STs). The cell proportions in organoids were in Matrigel CTs (44%), EVTs (33%), STs (22%) and in bioprinted PEG CTs (43%), EVTs (54%), STs (3%). They detected 2 subgroups with one of metal regulating CTs and one of maturing EVT by gene expression analysis (page 14). Interestingly, when they analyzed the protein expressions pattern in organoids from Matrigel and from bioprinted PEG, there were 33 proteins significantly upregulated in organoids from Matrigel, and 1238 proteins upregulated in bioprinted PEG. These proteins correlated to 2 pathways (the biosynthesis of ubiquinol and heme) for organoids from Matrigel and to 114 pathways for organoids from bioprinted PEG. At Day 5 of culture, the bioprinted organoids were subjected to TNF (20 ng/mL) to induce inflammation and treated at day 8 of culture with aspirin (0.5 mM) or metformin (0.5 mM). The results showed that both drugs induced the growth arrest of the organoids. Finally, 3 days old organoids were extracted from their matrix (Matrigel or PEG) layer and cultured in suspension for 12 or 28 days. They displayed a regular polarization, a spherical morphology, and the cavity at the core of the organoids increased with the organoid growth.

Positive:

The authors made a strong analysis on the formed organoids. They detailed the cell proportion (CTs, STs, EVT) in the organoids obtained with both matrices (Matrigel, printed PEG), which highlighted the materials effects on the organoid formation due to the marked differences in the results. They deepened their analysis by gene analysis and defined two additional cell subclusters (a metal regulating CTs subgroup, and a maturing EVT subgroup). Moreover, they used proteomic analysis and revealed important differences in protein expression and upregulated pathways between organoids formed in Matrigel and in printed PEG matrices.

Another important point is the organoids polarity. The manuscript showed by immuno-fluorescence staining that culturing trophoblast organoids in suspension (pages 22-24) induced the formation of STs layer on the outer, which correspond to a regular polarity.

It is also interesting to see that a bioprinter can be used to generate some placenta organoids knowing the speed, automation, resolution, precision, and easy to use of the technique.

Less positive:

The last part of the paper on drug testing is less informative. The inflammation on the organoids is not really defined. There are no immune cells, no concentration curve of TNF toward organoids (however the authors gave some results on organoid number, size, and cell viability (page 20) at the TNF concentration used (20 ng/mL) (page 30)), and no biological biomarkers that could give information on the status of inflammation on the organoids. The inflammation is suggested by the fact that women with preeclampsia have high level of TNF (page 20) and that TNF is usually a marker of inflammation. Secondly, if in the drugs tested to treat the inflammation the choice of aspirin is understandable (since it acts on the cyclooxygenase), the choice of metformin which is considered as an antidiabetic is more debatable. Third, the authors observed an impeded organoid growth when treated with one of these drugs. The organoid growth is of course one parameter that can be considered and certainly a sensitive parameter in the frame of organoids and toxicology, however, the impaired or growth arrest is not so much informative.

The title highlights the word "bioprinted", the study compares two different matrices with one bioprinted, and the authors claimed to have fabricated the first bioprinted placental organoids, which looks true. However, curiously the paper does not appear like a usual bioprinting paper with fabrication and characterization of a bioink and optimization of the bioprinting parameters. Although, the manuscript mentioned the bioprinter and the commercial bioink used (page 30) there are no bioprinting parameters given. The authors have just used a bioprinter. The paper strongly highlights the matrices, and the bioprinting is accessory.

Globally:

Globally, the paper is well written with a linear story. The figures are fine and informative supporting the text. The bibliography is correctly done with 42% of the references cited within the last five years.

Development of a model of placenta in vitro that mimic in vivo situation is a challenging task due to the dynamic, the complexity, and the number of different cells involved in the process in vivo. The first 3D models of placental organoids were

introduced by Turco et al. in 2018 and by Haider et al. 2018. These organoids were grown from primary trophoblasts isolated from first trimester placental tissue. Then, other placental organoids were generated using other primary tissues, induced pluripotent stem cells, and trophoblast cell lines (Dietrich et al. 2023, Mao et al. 2023, Sato et al. 2021). As the authors mentioned it, Matrigel was often used in these systems. However, in more recent papers some authors started to use other materials such as agarose and suspended cultures, or collagen IV beads and suspended cultures (Hori et al. 2024, Zhou et al. 2023).

The present paper strongly highlights the role of the matrix used when generating trophoblast organoids. This is innovative and the main message from this paper. This suggests testing a library of biomaterials used as a matrix to observe their effects on the trophoblast organoid formation and open opportunities to control the organoids formation in a planned way.

The second point is the polarity of the organoids. Standard cultures of human trophoblast organoids with embedded matrix often do not recapitulate the normal polarity of the organoids which present an inversed polarity with an inward facing STs and an outward facing CTs (Turco et al. 2018, Yang et al. 2022). Several studies have now shown that culturing the trophoblast organoids in suspension induced the normal polarization of the organoids with the STs on the outer surface (Yang et al. 2024, Zhou et al. 2023, Hori et al. 2024).

The present paper confirms the beneficial effect of culturing trophoblast organoids in suspension (pages 22-24) with the formation of STs layer on the outer.

The paper is interesting providing new insights on the development of placenta organoid model and contributing to the forefront of the research on this field. The paper is publishable after minor revisions.

Minor revisions

- Figure 6 page 23 is wrongly labeled fig 3. (see also extended data Fig. 6 not Fig 3)
- When you are talking about drugs TNF aspirin, metformin, it would be good to mention the concentrations used between bracket in the text. Notably page 20 (Organoid treatment) it would be good to add in the text what type of treatment you did rather than to give just the results.

Version 1:

Reviewer comments:

Reviewer #1

(Remarks to the Author)

The authors have made significant revisions to their manuscript. The manuscript is well written, and the logic/flow of the process is much more detailed and step-by-step. I feel the changes made, both to my comments and the other reviewer's comments, have strengthened the quality of the paper.

I commend the authors for performing in vivo cell type (single-cell) comparisons to their organoid model in bioprint and Matrigel matrix. I also commend the authors for investigating the utility of their PEG-based matrix for supporting hTSC formation. As is, the manuscript is markedly improved and is an important resource for the trophoblast research (and organoid-using) community.

A minor request, concerning the statement of placenta tissue stiffness of 1.1 KPa, Line 114, is to include a supporting reference for this statement.

Reviewer #2

(Remarks to the Author)

The authors have correctly answered the comments of the different reviewers and optimized their manuscript. I have no more comment and the manuscript is good and publishable.

Point-to-point responses to Reviewers' comments

We thank the Editor and Reviewers for their thorough review and constructive comments regarding our manuscript and for providing sufficient time for us to complete additional experiments and revise the manuscript. We have addressed all the comments which has significantly improved our manuscript, and it is now ready for publication in *Nature Communications* journal, but we would of course be open to considering any further recommended revisions.

Reviewer #1 (Remarks to the Author):

The manuscript by Richards et al set out to test organoid forming ability of the trophoblast/choriocarcinoma hybrid cell line in a synthetic bioprinted PEG scaffold. The rationale of the study centers on the valid concern on the reliance of the extracellular matrix substance, Matrigel. Derived from rodent sarcoma, Matrigel has significant batch variability and the composition of the proteins are not entirely well defined. This inconsistency has undoubtedly contributed to discrepancies in scientific findings. Therefore the underlying goal of the work to examine the feasibility of implementing a synthetic PEG-based scaffold for organoid culture is valid and important.

We thank the Reviewer for outlining the significance of our study for placental biology field and overall organoid culture. Our manuscript provides a comprehensive assessment of the organoid formation potential of first trimester trophoblast cells, the importance of matrix selection for more broad organoid model development and its application as a drug screening platform.

I had difficult time reviewing this manuscript as there seemed to be missing components in the results section. But as written, I had a difficult time following the logic of the specific extracellular matrix proteins selected for PEG bioprinting. Moreover, I was unsure which protein was selected for PEG bioprinting; based on the discussion it appears the authors performed most if not all their characterizations using the naked PEG scaffold. However, as indicated, this was not clear within the manuscript.

We apologise to the Reviewer for finding it difficult to navigate through our manuscript. We have gone through our manuscript in detail, substantially revised it and have improved the flow of the paper plus included further clarification related to the choice of bioinks/ECM used for bioprinting.

The bioinks are PEG-based with chemically bonded adhesion peptides (IKVAV, YIGSR, GFOGER) mimicking laminin- α and β and collagen I that form synthetic hydrogels from a cross-link reaction with an activator formulation (Ref 108). The rationale for choosing these adhesion peptides was to closely resemble human placental tissue microenvironment (Ref 90, 109-111). These were manufactured and provided by a company called Inventia Life Sciences in Australia without any further modification, thus reproducibility is ensured through the manufacturer's QC methods for any lab utilising the RASTRUM bioprinting platform. The matrix components are tailored to the placental microenvironment by matching the stiffness and ECM components of the placental tissue. The matrix stiffness of 1.1 kPa provides an initial scaffolding that will enable the cells to form tissue structures over time which may progressive

become stiffer beyond the matrix stiffness parameters provided. We have now added this information on Page 5 (Results) & 35 (Methods).

In addition to not understanding the flow/logic of the paper, I also have concerns related to the use of the ACH-3P cell line as a trophoblast organoid model. While not entirely novel (as the authors rightly indicate; Dietrich et al, Placenta 2023 PMID: 36696785), the authors chose to use a cell culture media that is somewhat ill-defined and that does not promote conditions of trophoblast regeneration or stemness. This decision limits how the current model can be directly compared to current state-of-art trophoblast organoid systems, and importantly, dampens the potential impact of the manuscript as an application to progenitor/stem cell-based trophoblast organoid systems.

We appreciated Reviewer's concerns and to address these, we have now completed additional data analysis and experiments with trophoblast stem cells using differentiation media to demonstrate further application of this 3D bioprinting methodology (new Figure 6). We chose ACH-3Ps because previous studies have shown that this cell line closely resembles the primary first trimester trophoblasts (Ref 106/107) and because our goal is to develop a low-cost, high-throughput biologically relevant model. This 3D bioprinted trophoblast organoid platform can accelerate drug discovery for placental dysfunction disorders such as preeclampsia and supplement the use of primary cells that require difficult-to-obtain placental samples and expensive culture conditions.

A key finding of our research was the capacity for extracellular matrix features to drive the differentiation of cells in a 3D model (in the absence of chemically induced trophoblast stem/progenitor cell differentiation). To more closely resemble native tissues, matrix composition and mechanical features must be closely considered. Further comparison of ACH-3P organoids to first trimester placental tissue and primary trophoblast organoids using publicly available transcriptomics data have now been included, strengthening our confidence in the similarity of ACH-3P organoids to the tissues they aim to reflect and the current gold-standard (Figure 3I). To assess the application of our organoid culturing method to widely accepted trophoblast stem cells (TSCs), we employed TSC line, CT29, from Riken biobank. Though TSC organoids formed in both Matrigel and 1.1kPa blank bioprinted PEG, they were significantly smaller in the stiffer, naked PEG-based matrix. We also trialled 'rich' PEG-based matrix of the same stiffness containing RGD (fibronectin, laminin), YIGSR (laminin) and CNYYSNS (collagen IV) adhesion peptides with laminin-511 and hyaluronic acid (Inventia Life Science, cat. Px02.72), described on page 35. While the 'rich' matrix increased TSC organoid size and number, they were still fewer and smaller compared to those in Matrigel (new Figure 6). This is likely due to TSCs' affinity for softer matrix conditions, resembling the microenvironment during decidualisation in preparation for blastocyst implantation. This highlighted the crucial role of matrix stiffness in supporting the placental stem cell niche. We have now discussed this in the manuscript (Page 32).

Below are my specific comments:

1. Figure 1 is present in the manuscript body, but prose in the results related to Figure 1 are missing.

Figure 1 was referred to in the Main text section (page 5), acting as a graphical abstract. However, we have now removed this and re-numbered the figures, which is aligned to *Nature Communications* author's guidelines.

2. Relatedly, there appears to be missing text related to the logic and choice of specific ECM proteins being incorporated into PEG hydrogels. As such, it was not possible to adjudicate the upfront justification of the use of naked PEG for all downstream analyses. This is a major limitation of the current manuscript as this is the major novel component of the work and likely the aspect of the work that could be highly valuable to the field.

We appreciate this comment and apologise for the lack of clarification. The choice of extracellular peptides in the initial matrix selection has now been justified in the Methods (page 35) and the Results section (Page 5) based on their expression in native placental tissue. The decision to proceed with the blank (naked) PEG matrix was based on an increased number of organoids formed in this matrix compared to other matrices containing α - and β -laminin or collagen I, and no significant impact on viability - this has now been explained in the Results section (Extended Data Figure 1a,c).

3. This reviewer really questions the use of the culture conditions chosen (Hams F12 with 10% FBS) along with the ACH-3P cell line. While the justification of using an immortalized cell line could be made, the authors didn't really make it clear why this particular cell line was chosen for organoid forming abilities. The cell line used spontaneously differentiates into SCT- and EVT-like cells. This is an issue for the utilization of this model for studying trophoblast differentiation as the system lacks control and the differentiation kinetics are somewhat random.

We thank the Reviewer for this comment and have clarified the decision to test the organoid forming potential of ACH-3P cells in the Methods section (page 33/34). The decision to test ACH-3P was based on three main reasons: (i) at the initiation of this study, there was no literature describing their organoid formation, (ii) ACH-3Ps are transcriptionally more similar to primary trophoblasts than most other immortalised trophoblast cell lines (Ref 107), (iii) they had demonstrated capacity to differentiate into at least two trophoblast subtypes in 2D culture and iv) they represented a low-cost, low-risk and high-throughput model that does not require expensive media components and therefore has a potential to accelerate drug discovery.

The capacity of ACH-3P organoids to resemble first trimester placental tissue (6 weeks and 8 weeks) and primary trophoblast organoids is now supported by our additional scRNAseq comparisons to published data (Figure 31). While trophoblast differentiation occurred within this simplified culture medium, we demonstrated that matrix composition alone could control differentiation. Though we have not tested trophoblast stem cell medium on ACH-3P cells, this could be used to maintain stemness and prevent differentiation if desired. However, reliance on chemical mediators can inhibit mechanistic studies of these differentiation processes so an alternate method of controlling differentiation is an advantage.

The capacity for ACH-3P organoids to be used as a biologically relevant low-cost, low-risk and reproducible model of early placental development is now highlighted in the Discussion (page 32).

4. Another limitation of the manuscript was the complete lack of comparison of ACH-3P data to *in vivo*. Single cell datasets of first trimester chorionic villi exist and would greatly improve the characterization of the ACH-3P model and importantly, the impact of different matrices on trophoblast biology. As such, it is difficult to adjudicate how the synthetic hydrogel impacts trophoblast regeneration and differentiation without comparison to a gold standard control.

We thank the Reviewer for this recommendation, which we have now taken on board and completed. As explained above, our scRNAseq datasets were compared to publicly available data from human 6 week and 8-week chorionic villi as well as primary first trimester cytotrophoblast derived organoids (page 14, Figure 3l & Extended Data Figure 2e-g). Our comparisons have demonstrated significant similarity between trophoblast subtypes identified in our samples and those of *in vivo* and *in vitro* first trimester tissue. It is worth noting that bioprinted organoids had stronger associations with the 6-week placental samples than Matrigel-embedded organoids. We have now included this in the discussion further supporting our 3D bioprinted ACH-3P trophoblast organoid model as representative of the first trimester placental tissue (Page 30).

5. Related to comment 3 above, I had a difficult time understanding why the authors did not examine how their PEG bioprint conditions impacted TSC organoid forming abilities, as TSC lines (Okae lines) are the current gold standard in trophoblast biology research due to their commercial availability and ease of use. While there are geographic locations that prohibit the use of TSCs, iPSC-derived TSCs (PMID: 35523141) and iTSCs (PMID: 32939092) bypass these restrictions. I feel the authors needed to make a strong case for focussing on the ACH-3P line, as it is arguably a lesser cell model for studying trophoblast biology.

We thank the Reviewer for this suggestion and have addressed their comment in two ways: as described above, we have justified the choice of ACH-3P line and have performed additional experiments to assess the application of our 3D bioprinted method using PEG-based matrices to TSC organoids. We used commercially available “Okae” TSC line CT29, initially derived from a male first trimester placenta by Okae and colleagues (Ref 69). TSC organoids formed in both Matrigel and bioprinted conditions using the same 1.1kPa naked PEG matrix and were able to be differentiated using chemical mediators (Page 24/25, Figure 6). Interestingly, less organoids formed from bioprinted TSCs and they were significantly smaller than their Matrigel counterparts. However, the decidua is prepared to receive the blastocyst containing an outer layer of TSCs by softening from the proliferative phase to the secretory phase of the menstrual cycle (Ref 86-87). Further, extracellular matrix protein expression changes throughout the menstrual cycle and during implantation at the site where the blastocyst adheres, which could be used to inform future matrix optimisation experiments for TSC organoid niche formation in softer synthetic hydrogels (Ref 89/90). This has now been discussed on Page 32.

Reviewer #2 (Remarks to the Author):

In this study, the authors 3D bioprinted trophoblast organoids and compared them with conventional Matrigel embedding trophoblast organoids. They used the trophoblast cell line ACH-3P encapsulated in poly(ethylene glycol) (PEG) bioink. They observed that ACH-3P in Matrigel or in PEG spontaneously formed organoids after 4 days of culture and these organoids were metabolically active and cells proliferation increased with time. The organoids were formed by cytotrophoblasts (CTs) with a central cavity, while the surface of the organoids was covered with extravillous trophoblasts (EVTs) with some clusters of syncytiotrophoblasts (STs). The cell proportions in organoids were in Matrigel CTs (44%), EVTs (33%), STs (22%) and in bioprinted PEG CTs (43%), EVTs (54%), STs (3%). They detected 2 subgroups with one of metal regulating CTs and one of maturing EVTs by gene expression analysis (page 14). Interestingly, when they analyzed the protein expressions pattern in organoids from Matrigel and from bioprinted PEG, there were 33 proteins significantly upregulated in organoids from Matrigel, and 1238 proteins upregulated in bioprinted PEG. These proteins correlated to 2 pathways (the biosynthesis of ubiquinol and heme) for organoids from Matrigel and to 114 pathways for organoids from bioprinted PEG.

At Day 5 of culture, the bioprinted organoids were subjected to $TNF\alpha$ (20 ng/mL) to induce inflammation and treated at day 8 of culture with aspirin (0.5 mM) or metformin (0.5 mM). The results showed that both drugs induced the growth arrest of the organoids. Finally, 3 days old organoids were extracted from their matrix (Matrigel or PEG) layer and cultured in suspension for 12 or 28 days. They displayed a regular polarization, a spherical morphology, and the cavity at the core of the organoids increased with the organoid growth.

We are very grateful to the Reviewer for taking the time to provide constructive feedback on our manuscript and for summarising the key findings elegantly.

Positive:

The authors made a strong analysis on the formed organoids. They detailed the cell proportion (CTs, STs, EVT) in the organoids obtained with both matrices (Matrigel, printed PEG), which highlighted the materials effects on the organoid formation due to the marked differences in the results. They deepened their analysis by gene analysis and defined two additional cell subclusters (a metal regulating CTs subgroup, and a maturing EVT) subgroup). Moreover, they used proteomic analysis and revealed important differences in protein expression and upregulated pathways between organoids formed in Matrigel and in printed PEG matrices.

Another important point is the organoids polarity. The manuscript showed by immunofluorescence staining that culturing trophoblast organoids in suspension (pages 22-24) induced the formation of STs layer on the outer, which correspond to a regular polarity. It is also interesting to see that a bioprinter can be used to generate some placenta organoids knowing the speed, automation, resolution, precision, and easy to use of the technique.

We thank the Reviewer for their careful consideration of our manuscript and for clearly outlining the strengths of our study. Indeed, this is the first report, to our knowledge, that describes the use of 3D bioprinting for trophoblast organoids that can accelerate drug discovery for placental dysfunction disorders such as preeclampsia and lead to better understanding of placental biology.

Less positive:

The last part of the paper on drug testing is less informative. The inflammation on the organoids is not really defined. There are no immune cells, no concentration curve of TNF α toward organoids (however the authors gave some results on organoid number, size, and cell viability (page 20) at the TNF α concentration used (20 ng/mL) (page 30)), and no biological biomarkers that could give information on the status of inflammation on the organoids. The inflammation is suggested by the fact that women with preeclampsia have high level of TNF α (page 20) and that TNF α is usually a marker of inflammation.

We thank the Reviewer for their insightful comments, which we have fully taken on board and addressed. We have now conducted additional experiments using a concentration curve of TNF α (1, 5, 20 ng/ml) against bioprinted organoids to further justify the selected concentration. This included organoid size, number, proliferation, viability and quantification of C reactive protein (CRP), a marker of inflammation, in organoid conditioned medium (Extended Data Figure, page 18). We showed that 20ng/ml dose led to the highest increase in media CRP concentration whereas all doses reduced 3D bioprinted trophoblast organoids proliferation. We have also quantified IL-6 gene expression within organoids treated with TNF- α (20ng/ml) \pm aspirin/metformin (Figure 3 extended data). We have now included further explanation of why TNF α , a key inflammatory cytokine, was used to emulate inflammation in preeclampsia in the Results (page 18, Ref 35).

Secondly, if in the drugs tested to treat the inflammation the choice of aspirin is understandable (since it acts on the cyclooxygenase), the choice of metformin which is considered as an antidiabetic is more debatable.

As we used TNF- α as a preeclamptic stimuli for 3D bioprinted trophoblast organoids, we wanted to screen established and emerging treatments such as aspirin and metformin, respectively. We have now added a paragraph justifying our choice of treatments on Page 4 and 19 with references (Refs 35-37). Metformin is emerging as a new treatment for preeclampsia that has also been tested in clinical trial settings in women with severe preeclampsia showing that it can extend gestational age of delivery by 7 days, which is an astonishing result (Ref 53).

Third, the authors observed an impeded organoid growth when treated with one of these drugs. The organoid growth is of course one parameter that can be considered and certainly a sensitive parameter in the frame of organoids and toxicology, however, the impaired or growth arrest is not so much informative.

We agree with the Reviewer that organoid growth is not the most useful parameter, and to address this comment we have now performed an array of assays to show the full capacity of parameters that our 3D bioprinted model can achieve including organoid number, organoid size, proliferation/metabolic activity, CRP concentration in the media, IL-6 gene expression within organoids, and cytotrophoblast differentiation to STBs and EVT s (Figure 4). Given we have performed a new full bioprint to determine different parameters which have enhanced our understanding of the impact of TNF- α \pm aspirin/metformin, we have replaced the whole figure with this new data (Page 19/20).

The title highlights the word “bioprinted”, the study compares two different matrices with one bioprinted, and the authors claimed to have fabricated the first bioprinted placental organoids, which looks true. However, curiously the paper does not appear like a usual bioprinting paper with fabrication and characterization of a bioink and optimization of the bioprinting parameters. Although, the manuscript mentioned the bioprinter and the commercial bioink used (page 30) there are no bioprinting parameters given. The authors have just used a bioprinter. The paper strongly highlights the matrices, and the bioprinting is accessory.

We appreciate the comment regarding our use of the bioprinter and commercially available matrices and have now provided additional information about the bioprinting process and bioinks. We have now added references describing the bioprinting parameters in the Methods (page 35) and Results (page 4/5). While the commercial bioink formulations can be changed, the platform does not allow users to adapt bioprinting parameters besides the choice of defined plug sizes, ensuring reproducibility through the manufacturer’s QC methods for any lab utilising the RASTRUM bioprinting platform. We have further highlighted the advantage of bioprinting to improving precision, reproducibility and scale-up potential in the Discussion (page 32).

Globally:

Globally, the paper is well written with a linear story. The figures are fine and informative supporting the text. The bibliography is correctly done with 42% of the references cited within the last five years.

We thank the Reviewer for their positive feedback on our manuscript.

Reviewer #3 (Remarks to the Author):

Development of a model of placenta in vitro that mimic in vivo situation is a challenging task due to the dynamic, the complexity, and the number of different cells involved in the process in vivo. The first 3D models of placental organoids were introduced by Turco et al. in 2018 and by Haider et al. 2018. These organoids were grown from primary trophoblasts isolated from first trimester placental tissue. Then, other placental organoids were generated using other primary tissues, induced pluripotent stem cells, and trophoblast cell lines (Dietrich et al. 2023, Mao et al. 2023, Sato et al. 2021). As the authors mentioned it, Matrigel was often used in these systems. However, in more recent papers some authors started to use other materials such as agarose and suspended cultures, or collagen IV beads and suspended cultures (Hori et al. 2024, Zhou et al. 2023).

The present paper strongly highlights the role of the matrix used when generating trophoblast organoids. This is innovative and the main message from this paper. This suggests testing a library of biomaterials used as a matrix to observe their effects on the trophoblast organoid formation and open opportunities to control the organoids formation in a planned way.

We thank the Reviewer for their careful consideration of our manuscript and providing constructive feedback in the context of the literature. Indeed, a key finding of our research was the capacity for extracellular matrix features to drive the differentiation of cells in 3D model (in the absence of chemically induced trophoblast stem/progenitor cell differentiation). We have now conducted further comparison of ACH-3P bioprinted organoids to first trimester placental tissue and primary trophoblast organoids using publicly available transcriptomics data, strengthening our confidence in the similarity of ACH-3P organoids to the tissues they aim to reflect and the current gold-standard (Figure 3I). This is the first report, to our knowledge, that describes the use of bioprinting for trophoblast organoids that can accelerate drug discovery for placental dysfunction disorders such as preeclampsia and lead to better understanding of placental biology.

The second point is the polarity of the organoids. Standard cultures of human trophoblast organoids with embedded matrix often do not recapitulate the normal polarity of the organoids which present an inversed polarity with an inward facing STs and an outward facing CTs (Turco et al. 2018, Yang et al. 2022). Several studies have now shown that culturing the trophoblast organoids in suspension induced the normal polarization of the organoids with the STs on the outer surface (Yang et al. 2024, Zhou et al. 2023, Hori et al. 2024). The present paper confirms the beneficial effect of culturing trophoblast organoids in suspension (pages 22-24) with the formation of STs layer on the outer.

We agree with the Reviewer that we are not the first ones to show that the trophoblast organoids in suspension induced the normal polarization of the organoids with the syncytiotrophoblasts on the outer surface, however we confirmed that this is the case with both our Matrigel-embedded and bioprinted organoids showing their application for recapitulating placental tissue architecture accurately in suspension.

The paper is interesting providing new insights on the development of placenta organoid model and contributing to the forefront of the research on this field. The paper is publishable after minor revisions.

Minor revisions

1. Figure 6 page 23 is wrongly labelled fig 3. (see also extended data Fig. 6 not Fig 3)

We apologise for this error and thank the Reviewer for pointing this out which has now been rectified. The extended data figures are labelled according to *Nature Communications* authors guidelines. Many thanks.

2. When you are talking about drugs TNF α , aspirin, metformin, it would be good to mention the concentrations used between bracket in the text. Notably page 20 (Organoid treatment) it would be good to add in the text what type of treatment you did rather than to give just the results.

We thank the Reviewer for this suggestion and have added drug concentrations to the Results section (page 19).

Point-to-point responses to Reviewers' comments

We sincerely thank the Editor and Reviewers for accepting our manuscript for publication in *Nature Communications* journal. We have now completed all the editorial requests (Author's checklist attached), provided a featured image and addressed the remaining comment from the Reviewer #1.

Reviewer #1 (Remarks to the Author):

A minor request, concerning the statement of placenta tissue stiffness of 1.1 KPa, Line 114, is to include a supporting reference for this statement.

We thank the Reviewer for their suggestion. We have now added a reference (26) to the sentence stating that *“The matrix components are tailored to the placental microenvironment by matching the stiffness and extracellular matrix (ECM) components of the placental tissue and underlying decidua basalis”*.